# AgentSteerTTS: A Multi-Agent Closed-Loop Framework for Composite-Instruction Text-to-Speech

Bin Kang[1,2,3]    Shaoguo Wen[3]    Yang Fan[2]    Shunlong Wu[4]
Junjie Wang[2]    Yulin Li[2]    Junzhi Zhao[5]    Junle Wang[3]    Zhuotao Tian[2†]

[1]University of Chinese Academy of Sciences    [2]Shenzhen Loop Area Institute
[3]Tencent Turinglab    [4]Tsinghua University    [5]Southwest Jiaotong University

## Abstract

While existing text-to-speech (TTS) models exhibit high expressiveness, fine-grained control over composite instructions remains challenging due to the structural mismatch between discrete textual intents and continuous acoustic realizations. Inspired by human cognitive decoupling, we introduce **AgentSteerTTS**, a multi-agent closed-loop framework designed for intent-faithful expressive control of composite instructions. First, in our framework, an adversarial disentanglement agent mitigates speaker-emotion leakage through gradient reversal and cross-covariance regularization. Next, a Dual-Stream Anchoring Controller grounds abstract intents using a large-scale acoustic prototype library: a Retrieval Agent selects expressive anchors, while a Synthesis Agent fuses them into continuous control vectors via gated attention. Finally, a Fast–Slow Feedback Agent refines output intensity through latent gradient correction and resolves semantic–acoustic mismatches using high-level perceptual critique. Experiments on a composite-instruction benchmark and public test sets show that **AgentSteerTTS** yields consistent and significant improvements to the baselines, demonstrating the effectiveness of the proposed method. Project page: https://kane2kang.github.io/AgentSteerTTS/

†Corresponding Author: Zhuotao Tian
(zhuotaotian@slai.edu.cn).

*Proceedings of the 43rd International Conference on Machine Learning*, Seoul, South Korea. PMLR 306, 2026. Copyright 2026 by the author(s).

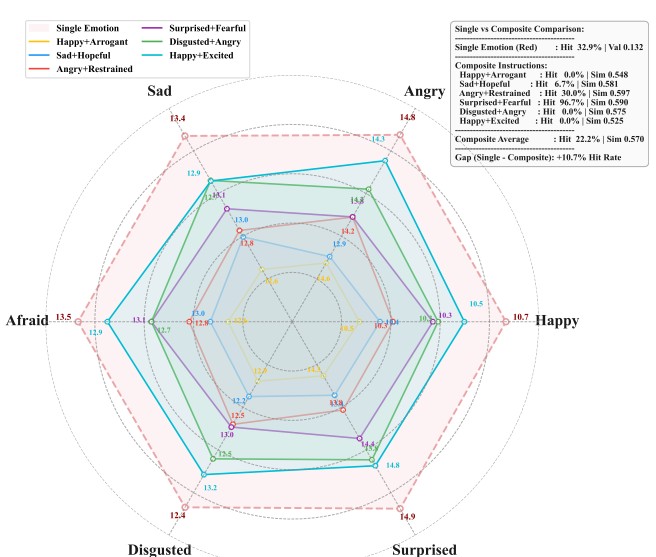

*Figure 1.* Illustration of Semantic-Acoustic Misalignment. The red dashed line delineates the target emotion, while the colored regions represent the generated speech. Despite high-level control instructions, the model exhibits under-expression along the target emotion dimension, while unintentionally leaking acoustic energy into irrelevant emotional dimensions.

## 1. Introduction

Recent advances in speech and multimodal foundation models (Li et al., 2025b; 2023; Kang et al., 2025; Li et al., 2026a;b) have significantly improved Text-to-Speech (TTS) synthesis, yielding more natural and fluent outputs with fewer artifacts. However, these gains have not been matched by comparable progress in expressive control. State-of-the-art TTS systems (Zhang et al., 2023; Rubenstein et al., 2023; Chu et al., 2024) often generate clear and natural speech but struggle to faithfully render user-specified emotional or stylistic attributes—particularly under composite instructions involving multiple, potentially conflicting constraints, such as "Happy but slightly Arrogant".

A key bottleneck stems from a common formulation (Cui

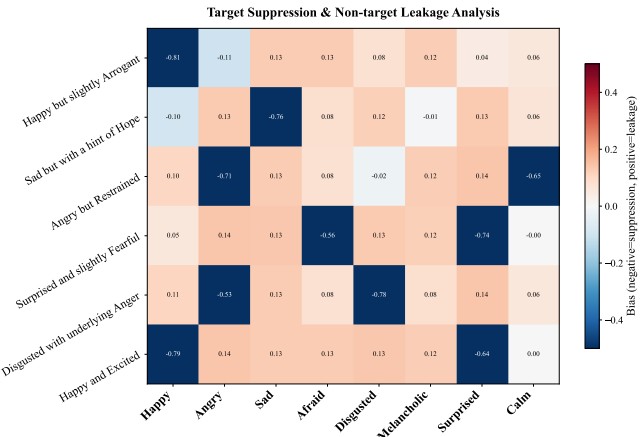

*Figure 2.* Emotion expression bias. Target emotion dimensions are suppressed, while non-target dimensions exhibit consistent positive leakage across conditions.

et al., 2025; Ju et al., 2024; Gao et al., 2025a) that casts expressive TTS as a feed-forward mapping from text (optionally augmented with style prompts) to waveforms. Many such methods rely on fine-grained supervision, such as dense prosody annotations or meticulously curated datasets (Chen et al., 2025b; Huang et al., 2025), which hinders scalability. This dependence incurs high annotation costs, limits generalization across speakers, and fails to capture the nuanced intensity and stylistic variations implicitly encoded in natural-language instructions (Gao et al., 2025a; Yang et al., 2025). Consequently, users are often forced into a trial-and-error loop of prompt tuning and resampling to approximate a desired composite expressive target (Huang et al., 2025).

**Key Observations.** To investigate the causes of this instability, we conducted a pilot study using an expressive TTS model (Zhou et al., 2025a) conditioned on composite prompts specifying multiple fine-grained attributes. As shown in Fig. 1, while the generated speech exhibits high naturalness, intent faithfulness remains unreliable: across six composite intents, the generated speech is weakly aligned with the target composite in an emotion-embedding space (mean cosine similarity $\approx 0.58$), and the joint satisfaction rate, defined as the fraction of samples where both the primary and secondary attributes exceed a fixed recognition threshold, is only $\sim 30\%$, with some intents collapsing to near-zero success.

Crucially, these failures are not due to random variation but reflect systematic, directional biases. Fig. 2 shows that target dimensions in the composite profile are consistently under-activated (relative drop $25\%$–$45\%$), while non-target dimensions receive spurious activation (average $\approx +0.08$), indicating structured attribute dilution rather than noise: *the model under-expresses specified attributes and redistributes their intensity into irrelevant acoustic dimensions,*

*producing fluent yet semantically misaligned outputs.* Therefore, achieving reliable composite control requires explicit alignment between high-level, multi-dimensional intent constraints and low-level acoustic realizations to mitigate this structural semantic–acoustic misalignment.

**Our Solution.** Motivated by this insight, we propose AGENTSTEERTTS, a multi-agent closed-loop framework for composite-instruction expressive steering. First, we introduce an Adversarial Disentanglement Module (ADM) that uses adversarial learning (Kim et al., 2021) to reduce speaker–emotion leakage and encourage latent factorization of expressive attributes. Second, we integrate a Dual-Stream Anchoring Controller (DAC), where a retrieval module fetches high-expressivity acoustic prototypes from a large reference library with perceptual pruning, and a synthesis controller maps discrete textual intents into continuous control vectors via gated fusion. Finally, we implement a Fast–Slow Feedback module that combines a differentiable Latent Consistency Predictor for rapid intensity calibration with a supervisor module for high-level perceptual guidance. This hierarchical design explicitly addresses semantic–acoustic misalignment across diverse speakers and unconstrained prompts, enabling fine-grained control over subtle composite expressions and emotional nuances. Extensive experiments on our composite-instruction benchmark and public test sets (Zhou et al., 2021) show that AGENTSTEERTTS consistently improves intent alignment while preserving speaker identity and speech naturalness. Generated speech demos are provided in the supplementary material. Our contributions are three-fold:

- We identify a systematic semantic–acoustic mismatch in composite-instruction TTS: under discrete composite prompting, specified fine-grained target attributes are under-expressed while non-target attributes leak.

- We propose AGENTSTEERTTS, a multi-agent closed-loop framework in which an Adversarial Disentanglement Agent reduces speaker–emotion leakage, a Dual-Stream Anchoring Controller grounds acoustic control using composite instructions and reference prototypes, and a Fast–Slow Feedback Agent further mitigates semantic–acoustic mismatches at both the latent and waveform levels.

- Experiments on our composite-instruction benchmark and public test sets show improved intent alignment while preserving speaker similarity, e.g., compared to the strongest baseline, E-SIM increases from 0.864 to 0.955 and S-SIM from 0.823 to 0.841.

## 2. Beyond Discrete Composite Instruction

This section presents a pilot study and principled analysis demonstrating why discrete text instructions fundamen-

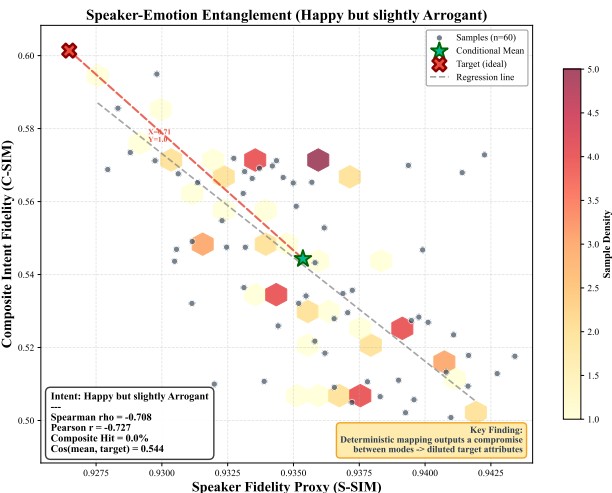

*Figure 3.* Speaker–emotion entanglement: Composite-intent fidelity (C-SIM) is negatively correlated with the speaker-fidelity proxy (S-SIM).

tally fail to reliably control continuous acoustic realizations for fine-grained composite instructions (e.g., "Happy but slightly Arrogant").

### 2.1. Why do deterministic mappings fail for composite instructions control?

We model speech generation as a stochastic process jointly determined by speaker identity $s \in \mathcal{S}$, linguistic content $\ell \in \mathcal{L}$, and continuous acoustic control variables $a \in \mathbb{R}^d$: $x \sim p(x \mid s, \ell, a)$, where $a$ governs expressive attributes such as prosody and emotion. A user instruction $t$ provides a set of semantic constraints $\mathcal{C}(t) = \{c_k\}_{k=1}^K$ (e.g., "Happy", "Arrogant"). Unlike a single label, a composite instruction does not specify a unique $a$; instead, it induces a feasible region in control space:

$$\mathcal{A}(t, s, \ell) \triangleq \{a \in \mathbb{R}^d \mid \forall c_k \in \mathcal{C}(t), \ \phi_k(a) \geq \tau_k\}, \quad (1)$$

where $\phi_k$ is an attribute (e.g., emotion) predictor and $\tau_k$ is a satisfaction threshold. For composite instructions, $\mathcal{A}(t, s, \ell)$ is often non-convex and non-trivial, implying multiple valid acoustic realizations.

However, many controllable TTS systems learn a single-point mapping from instruction to acoustics, $f_\theta(t; s, \ell) \mapsto \hat{a}$. This creates a key limitation: composite instructions admit many valid realizations (e.g., different combinations of speaking rate, pauses, and energy), so the conditional target distribution $p(a \mid t)$ is typically multi-modal (Le et al., 2023). A approximation is a mixture of $M$ Gaussians:

$$p(a \mid t) \approx \sum_{k=1}^M \pi_k(t) \mathcal{N}(a; \mu_k(t), \Sigma_k(t)), \quad (2)$$

where $\mu_k$ and $\pi_k$ are the mode center and weight. Under common regression objectives, a deterministic predictor

tends to return a mode-averaged estimate, $\hat{a} \approx \sum_k \pi_k \mu_k$. When modes correspond to distinct prosody patterns, this estimate often lies between modes (e.g., between "Happy-fast-Arrogant" and "Happy-slow-Arrogant"), which is perceived as weakened expression and attribute dilution. This may help explain why strong expressive TTS models can miss composite prompts like "Happy but slightly Arrogant": the output may drift toward a more neutral "Happy", or over-emphasize "Arrogant", instead of matching the intended combination.

### 2.2. Speaker–Prosody Entanglement

Audio generation faces another challenge: speaker timbre and emotion-related prosody are coupled in natural speech (Qu et al., 2025). Formally, let $s$ and $e$ denote speaker identity and the target composite prosody/emotion vector, respectively. In a latent-variable generative model:

$$z \sim p(z \mid s, e), \qquad x \sim p(x \mid z), \quad (3)$$

Under the real data distribution, the generation process is generally coupled,

$$p(z \mid s, e) \not\approx p(z \mid s), \qquad p(z \mid s, e) \not\approx p(z \mid e), \quad (4)$$

This indicates that timbre and prosody are not disentangled: they share latent acoustic cues (e.g., formants and spectral tilt). As a result, the realization of the same target $e$ can shift systematically across speakers $s$.

This coupling becomes especially harmful for composite instructions. Joint targets impose stricter constraints, making the identity–prosody trade-off more pronounced. When a model is required to satisfy a joint target $(s^\star, e^\star)$, generation effectively must trades off between speaker and prosody objectives:

$$\min_x \ \mathcal{L}_{\mathrm{spk}}\Big(\phi_{\mathrm{spk}}(x), \phi_{\mathrm{spk}}(x_{s^\star})\Big) + \lambda \, \mathcal{L}_{\mathrm{emo}}\Big(\phi_{\mathrm{emo}}(x), e^\star\Big), \quad (5)$$

Because identity- and prosody-related cues are correlated in acoustic space, composite-intent fidelity and an identity/timbre-consistency proxy can exhibit a negative association on real generations (Fig. 3), so models may "trade timbre for prosody" or "trade prosody for timbre," leading to instability and bias under composite control.

## 3. Method

This section introduces the proposed AGENTSTEERTTS. The architecture is designed to address the challenges of feature entanglement and fine-grained emotional control through a systematic decouple–anchor–refine mechanism. The overview of AGENTSTEERTTS is presented in Fig. 4.

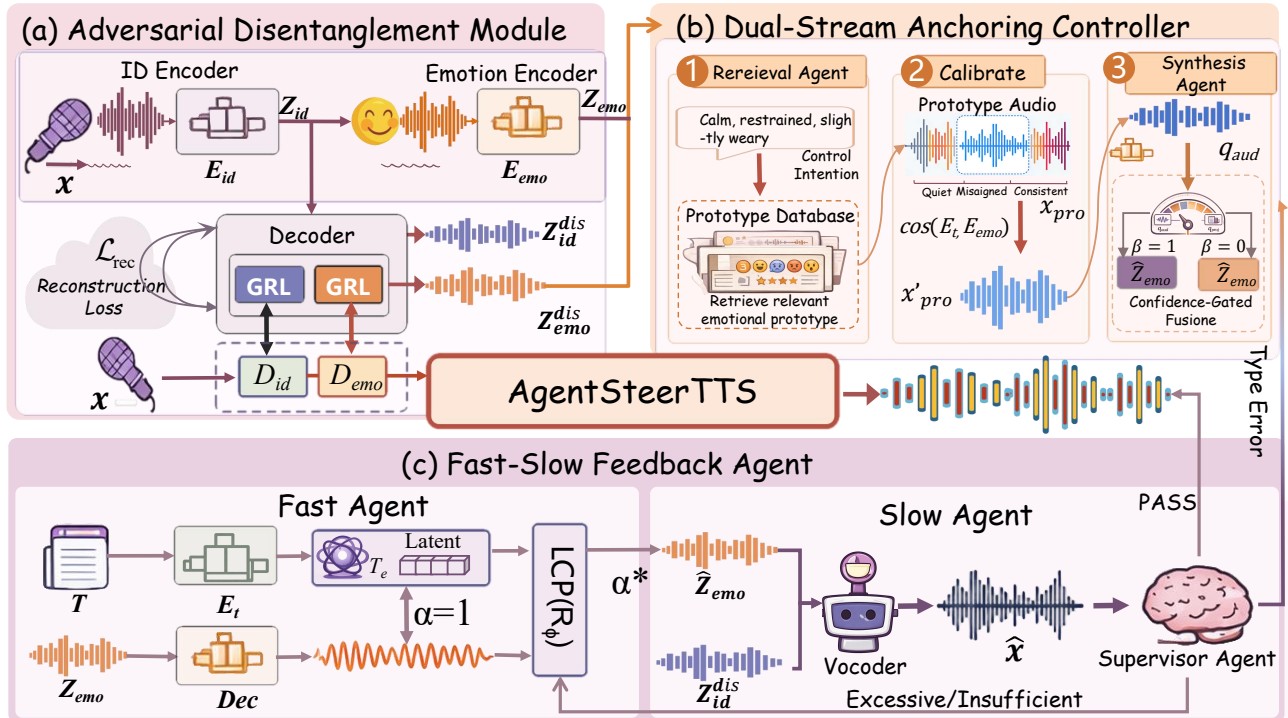

*Figure 4.* **Overview of the AGENTSTEERTTS architecture.** (a) an Adversarial Disentanglement Module that utilizes gradient reversal layers to construct orthogonal speaker and emotion subspaces; (b) a Dual-Stream Anchoring Controller that aligns latent features with retrieved acoustic prototypes via consistency calibration; and (c) a Fast-Slow Feedback Loop that dynamically refines the generation process during inference.

Concretely, ADM first makes the latent factors more separable, DAC then uses acoustic prototypes to place composite intents in the emotion subspace, and the Fast–Slow Feedback Agent further calibrates intensity and corrects residual semantic drift at inference time. In this paper, the term "agent" denotes a specialized functional component with its own objective, state, and decision rule inside the closed-loop generation process.

### 3.1. Adversarial Disentanglement Module

Current approaches often rely on high-level semantic control signals (e.g., emotion descriptions) and treat them as sufficient for stable steering. However, in acoustic space, speaker identity and emotion-related prosody are coupled across spectral and temporal patterns (Cho et al., 2025b;a). As a result, a single latent stream may mix identity traits with prosody cues, causing timbre drift or emotion leakage. To mitigate this issue, we propose an Adversarial Disentanglement Agent that factorizes a reference audio into two latent subspaces, $Z_{id}$ and $Z_{emo}$, with reduced cross-information. Our goal is targeted speaker–emotion decoupling for composite control rather than full factorization of all acoustic factors such as environment, accent, or age-related variation.

Specifically, for an input audio sample $x \in X$, we use an identity encoder $E_{id}$ and emotion encoder $E_{emo}$ to obtain utterance-level latents (via attention pooling):

$$z_{id} = E_{id}(x) \in \mathbb{R}^{d_{id}}, \quad z_{emo} = E_{emo}(x) \in \mathbb{R}^{d_{emo}}, \quad (6)$$

We expect $z_{id}$ to capture time-invariant identity and $z_{emo}$ to capture prosody-related cues. We apply a reconstruction objective so that their concatenation can recover the original acoustic features:

$$\mathcal{L}_{rec} = \left\| \mathrm{Dec}([z_{id}; z_{emo}]) - \mathrm{Mel}(x) \right\|_1, \quad (7)$$

where $\mathrm{Dec}(\cdot)$ is the feature decoder and $\mathrm{Mel}(x)$ is the Mel-spectrogram of $x$. Reconstruction alone can still admit degenerate solutions, allowing redundant information to leak into both subspaces. We therefore penalize the batch-wise cross-covariance between the two factors:

$$\mathcal{L}_{orth} = \left\| \mathrm{Cov}(z_{id}, z_{emo}) \right\|_F^2, \quad (8)$$

where $\mathrm{Cov}(\cdot, \cdot)$ denotes the cross-covariance computed over a mini-batch after mean-centering (see Appendix B.2).

To further suppress leakage, we adopt bidirectional adversarial training with two auxiliary discriminators, $D_{id}$ and $D_{emo}$, which predict speaker labels from the emotion stream and emotion labels from the identity stream, respectively:

$$\mathcal{L}_{adv} = \mathbb{E}\Big[ \mathrm{CE}\big(D_{id}(\mathrm{GRL}(z_{emo})), y_{id}\big)$$
$$+ \mathrm{CE}\big(D_{emo}(\mathrm{GRL}(z_{id})), y_{emo}\big)\Big], \quad (9)$$

where $y_{id}$ and $y_{emo}$ are ground-truth speaker and emotion labels, $\text{CE}(\cdot, \cdot)$ is cross-entropy, and $\text{GRL}(\cdot)$ is the Gradient Reversal Layer. During training, $D_{id}$ and $D_{emo}$ minimize $\mathcal{L}_{adv}$, while the GRL reverses gradients to $E_{id}$ and $E_{emo}$, encouraging $z_{id}$ to be emotion-invariant and $z_{emo}$ to be identity-invariant.

Overall, the disentanglement agent is trained with

$$\mathcal{L}_{\text{ADM}} = \mathcal{L}_{rec} + \lambda_{adv}\mathcal{L}_{adv} + \lambda_{orth}\mathcal{L}_{orth}, \quad (10)$$

which reduces speaker–prosody entanglement and stabilizes downstream emotional control.

### 3.2. Dual-stream Anchoring Controller

Once we obtain a more disentangled latent space, the next challenge is to reliably locate the target region in $Z_{emo}$. Using text labels alone often drifts toward neutral prosody, partly due to ambiguous text–emotion mappings. We therefore introduce the Dual-stream Anchoring Controller (DAC), which anchors generation with both explicit intent constraints and retrieved acoustic prototypes.

**Intent-Driven Retrieval.** To provide concrete acoustic priors, we build an emotion prototype library $\mathcal{M} = \{p_k\}_{k=1}^{N}$ with high-expressivity samples. Each prototype is paired with a fine-grained MLLM description $\tilde{t}_k$ and filtered by human listening to improve label quality and expressivity coverage.

Given a user instruction $t$, the Retrieval Agent rewrites $t$ into a small set of cue-focused queries $\{t^{(i)}\}_{i=1}^{M}$ and performs embedding-based search (with text encoder $E_{txt}$):

$$x_{pro} = \arg \max_{p_k \in \mathcal{M}} \max_{i \in [1,M]} \cos\left(E_{txt}(\tilde{t}_k), E_{txt}(t^{(i)})\right), \quad (11)$$

Simultaneously, a General Controller Agent parses $t$ into a discrete intensity vector $\mathbf{w} \in \mathbb{R}^{K}$[1]. We normalize/clamp $\mathbf{w}$ such that $\sum_{k=1}^{K} w_k \leq 1$.

**Consistency Calibration.** The retrieved prototype $x_{pro}$ may include irrelevant segments (e.g., silence or off-target emotions). We optionally refine it by selecting the temporal slice $x'_{pro}$ most consistent with the target intent:

$$x'_{pro} = \arg \min_{\tau \in x_{pro}} \mathcal{L}_{cos}\left(E_{emo}(\tau), E_{txt}(t)\right), \quad (12)$$

where $\tau$ denotes a sliding window within $x_{pro}$.

**Dual-stream Feature Extraction.** The controller converts the calibrated audio anchor $x'_{pro}$ and intent weights $\mathbf{w}$ into two complementary control embeddings:

---

[1]Following Ekman's theory of basic emotions (Ekman, 1992), which identifies six fundamental emotions and treats more complex affects as their combinations.

*(1) Acoustic Stream:* We encode the prototype and speaker reference audio into the emotion space as $q_{ref} = E_{emo}(x'_{pro})$ and $q_{base} = E_{emo}(x_{spk})$, where $E_{emo}$ denotes a lightweight W2V-BERT-based emotion encoder.

*(2) Text Stream:* To match the target speaker's voice, we use a speaker-adaptive lookup that selects style-consistent prototypes for each emotion category and aggregates them according to the intent weights $\mathbf{w}$, producing the symbolic control embedding $q_{txt}$.

**Adaptive Fusion.** To robustly integrate both streams, we employ a two-stage fusion. First, we create a continuous acoustic baseline by interpolating the reference speaker's neutral state $q_{base} = E_{emo}(x_{spk})$ with the retrieved anchor $q_{ref}$ using a factor $\lambda$:

$$q_{mix} = (1 - \lambda) \cdot q_{base} + \lambda \cdot q_{ref}. \quad (13)$$

Finally, we fuse the symbolic and acoustic controls. The text embedding $q_{txt}$ provides explicit intent guidance, while the remaining probability mass is assigned to the acoustic mix to maintain naturalness:

$$\hat{z}_{emo} = q_{txt} + \left(1 - \sum_{k=1}^{K} w_k\right) \cdot q_{mix}, \quad (14)$$

### 3.3. Fast-Slow Feedback Agent

Although the DAC provides high-quality initial features $\hat{z}_{emo}$, inference-time stochasticity can still cause semantic drift. Inspired by fast and slow thinking (Madaan et al., 2023), we introduce a Fast–Slow Feedback Agent with a Fast Agent (latent gradient correction) and a Slow Agent (perceptual semantic refinement).

**Fast Agent.** For efficient correction, inspired by efficient reasoning and compact intermediate representations (Li et al., 2026b;a), we introduce a lightweight, differentiable Latent Consistency Predictor (LCP), denoted by $R_\phi$, before the decoder. We use a scalar $\alpha \in [0, 1]$ to scale the injection of $\hat{z}_{emo}$. Before waveform synthesis, the LCP predicts an embedding $e_\phi(\alpha)$ from an intermediate Mel state and matches it to the target vector $v_{target} = E_{txt}(t)$:

$$e_\phi(\alpha) = R_\phi\left(\text{Dec}\left(z_{id}, \alpha \cdot \hat{z}_{emo}\right)\right). \quad (15)$$

Starting from $\alpha = 1.0$, we update $\alpha$ online by gradient descent:

$$\alpha^* = \alpha - \gamma \nabla_\alpha \mathcal{L}_c\left(e_\phi(\alpha), v_{target}\right), \quad (16)$$

where $\gamma$ is the step size and $\mathcal{L}_c$ is cosine distance(The details in Appendix B.1). This steers generation by adjusting latent scaling in Mel space, without running the vocoder.

**Slow Agent.** With the calibrated parameter $\alpha^*$, the system generates the final waveform $\hat{x}$ via a Synthesis Agent:

$$\hat{m}^* = \text{Dec}\left(z_{id}, \alpha^* \cdot \hat{z}_{emo}\right), \hat{x} = V(\hat{m}^*). \quad (17)$$

*Table 1.* Emotional expressiveness comparison on ESD. "Gemini3" and "Qwen3" denote different slow-loop critique backends within the same AGENTSTEERTTS pipeline, not standalone TTS baselines.

| Method | Control | WER ↓ | CER ↓ | MCD ↓ | SECS ↑ | ESMOS ↑ | SNMOS ↑ | SSMOS ↑ |
|---|---|---|---|---|---|---|---|---|
| Ground Truth | – | 6.98 | 2.94 | – | – | 4.57 (±0.06) | 4.61 (±0.06) | 4.24 (±0.07) |
| Emotional-TTS (Zhao & Yang, 2023) | Label | 34.38 | 24.30 | 10.916 | 0.760 | 2.59 (±0.09) | 2.19 (±0.09) | 2.79 (±0.11) |
| UMETTS (Tacotron) | Prompt | 24.42 | 14.12 | 10.684 | 0.780 | 2.21 (±0.21) | 2.80 (±0.09) | 3.18 (±0.09) |
| EmoSpeech (Ju et al., 2023) | Label | 7.91 | 3.52 | 8.040 | 0.841 | 4.22 (±0.07) | 4.03 (±0.08) | 3.87 (±0.06) |
| GenerSpeech (Zhao et al., 2022) | Audio | 12.30 | 6.74 | 6.812 | 0.840 | 4.01 (±0.09) | 3.98 (±0.07) | 3.66 (±0.08) |
| UMETTS (FastSpeech) | Prompt | 7.35 | 3.07 | 6.703 | 0.896 | 4.37 (±0.07) | 4.29 (±0.06) | 4.13 (±0.07) |
| **AgentSteerTTS (Gemini3 backend)** | **Prompt** | **7.25** | **3.21** | **5.942** | **0.854** | **4.35 (±0.06)** | **4.45 (±0.07)** | **4.25 (±0.08)** |
| **AgentSteerTTS (Qwen3 backend)** | **Prompt** | **7.12** | **3.08** | **5.815** | **0.861** | **4.42 (±0.05)** | **4.52 (±0.06)** | **4.31 (±0.07)** |

where $V(\cdot)$ is the vocoder. A Supervisor Agent then evaluates $\hat{x}$ and produces a natural-language critique $c_{critique}$ for fine-grained prosody (e.g., speaking rate, pauses, and intensity). If it detects a deviation, it triggers control updates:

- **Intensity Fine-tuning:** If the deviation is "Emotion too weak/strong," update $\alpha \leftarrow f(\alpha, c_{critique})$ and re-trigger the Fast Loop calibration.

- **Condition Reset:** If the deviation is "Incorrect emotion type," trigger DAC to re-retrieve and update $\hat{z}_{emo}$.

Overall, the Fast Agent provides differentiable intensity calibration before vocoding, while the Slow Agent performs waveform-level semantic checking and triggers either re-calibration or re-anchoring when mismatches persist.

## 4. Experiments

### 4.1. Implementation Details

**Datasets.** We fine-tuned our model using approximately 700 hours of emotional speech aggregated from ESD (Zhou et al., 2021) and MSP-Podcast (Busso et al., 2025). To construct the acoustic prototype library $\mathcal{M}$, we filtered 100 hours of high-expressivity samples with refined MLLM annotations, all resampled to 24kHz. We evaluate on 45,022 utterances (43.25 hours) across 7 emotion categories and 209 speakers: (1) ESD Benchmark (29.07 hours, 20 speakers, 5 emotions); and (2) Fine-grained Composite Dataset (14.18 hours) with composite instructions and subtle prosodic variations.

**Metrics.** We assess three dimensions: (1) Generation Quality: WER and QMOS, where QMOS denotes mean-opinion-score naturalness/overall quality. (2) Timbre Consistency: S-SIM computed by cosine similarity between speaker embeddings of synthesized and reference audio; on ESD we additionally report SECS as the speaker-embedding cosine similarity score. (3) Emotional Control: objective E-SIM using emotion2vec (Ma et al., 2024); on the composite benchmark, DMOS denotes instruction-following degree MOS,

SMOS denotes speaker-similarity MOS, and PMOS denotes perceived emotion-intensity MOS; on ESD, ESMOS denotes emotion-style similarity MOS, SNMOS denotes speech naturalness MOS, and SSMOS denotes speaker-similarity MOS.

**Implementation setup.** We fine-tune on ∼700h emotional speech (ESD (Zhou et al., 2021), MSP-Podcast (Busso et al., 2025)) and build a prototype library $\mathcal{M}$ by curating 100h high-expressivity clips from this pool with refined MLLM annotations (24kHz). We evaluate on 45,022 utterances (43.25h) across 7 emotions and 209 speakers: ESD (29.07h, 20 speakers, 5 emotions) and a 14.18h fine-grained composite set. Metrics cover quality (WER and task-specific MOS variants), timbre (S-SIM/SECS), and emotion control (E-SIM via emotion2vec (Ma et al., 2024)). We train with AdamW ($2 \times 10^{-4}$, 3k warmup) on 8×H20; the Slow Loop adds ≤200ms.

### 4.2. Main Results

**Emotional Expressiveness Comparison.** Table 1 reports emotional expressiveness on ESD. Among synthesized systems, AGENTSTEERTTS attains the highest ESMOS (4.42), SNMOS (4.52), and SSMOS (4.31) with the Qwen3 slow-loop backend. It also achieves the lowest MCD (5.815) while maintaining competitive intelligibility (WER=7.12), close to Ground Truth (6.98). We emphasize that "Gemini3" and "Qwen3" here denote backend replacements inside the same AGENTSTEERTTS pipeline rather than standalone TTS baselines; the small gap between the two settings suggests that the main gain comes from our decouple–anchor–closed-loop design rather than from the evaluator alone.

**Composite Instruction Comparison.** Table 2 reports results on our composite-instruction benchmark. AGENTSTEERTTS improves composite alignment, reaching E-SIM=0.955 (Gemini3 backend) vs. IndexTTS2 (0.864) and CosyVoice2 (0.748), and achieves the highest speaker similarity (S-SIM=0.841). It also attains the lowest WER (1.34) with stable subjective quality under strict emotion constraints (SMOS=4.28, PMOS=4.40, QMOS=4.38).

*Table 2.* Comparison on composite-instruction control and speech quality metrics. Our full system is retrieval-augmented; therefore, the comparison should be interpreted as composite-control performance rather than a strict retrieval-free zero-shot comparison.

| Category | Method | S-SIM ↑ | WER (%) ↓ | E-SIM ↑ | DMOS ↑ | SMOS ↑ | PMOS ↑ | QMOS ↑ |
|---|---|---|---|---|---|---|---|---|
| **Flow/Open** | VALL-E (Chen et al., 2025b) | 0.712 | 4.25 | 0.685 | 3.52±0.21 | 3.52±0.21 | 3.41±0.25 | 3.58±0.15 |
| | F5-TTS (Chen et al., 2025c) | 0.795 | 2.14 | 0.715 | 4.05±0.15 | 4.05±0.15 | 3.92±0.18 | 4.15±0.11 |
| | CosyVoice (Du et al., 2024a) | 0.812 | 1.95 | 0.732 | 4.18±0.12 | 4.18±0.12 | 3.98±0.16 | 4.21±0.10 |
| | CosyVoice2 (Du et al., 2024b) | 0.825 | 1.88 | 0.748 | 4.31±0.11 | 4.31±0.11 | 4.15±0.14 | 4.35±0.09 |
| **LLM-TTS** | Spark-TTS (Wang et al., 2025d) | 0.788 | 2.32 | 0.724 | 4.12±0.14 | 4.12±0.14 | 4.05±0.19 | 4.12±0.12 |
| | IndexTTS | 0.804 | 2.10 | 0.842 | 4.15±0.18 | 4.15±0.18 | 4.28±0.15 | 4.08±0.11 |
| | IndexTTS2 | 0.823 | 1.81 | 0.864 | 4.24±0.19 | 4.24±0.19 | 4.42±0.12 | 4.15±0.10 |
| **Ours** | AgentSteerTTS (Gemini3 backend) | **0.841** | 1.34 | **0.955** | 3.82±0.13 | 4.28±0.12 | 4.40±0.13 | 4.38±0.07 |
| | AgentSteerTTS (Qwen3 backend) | 0.817 | 1.57 | 0.921 | 3.83±0.19 | 4.26±0.13 | 4.36±0.10 | 4.36±0.08 |

*Table 3.* Ablation on composite-instruction control. We additionally report CSR (composite success rate), non-target emotion leakage, and timbre drift ∆S-SIM.

| Setting | S-SIM ↑ | ∆S-SIM ↓ | E-SIM ↑ | CSR ↑ | Leakage ↓ | WER ↓ | QMOS ↑ |
|---|---|---|---|---|---|---|---|
| **Full: AgentSteerTTS** | **0.841** | **0.021** | **0.955** | **0.78** | **0.14** | **1.34** | **4.38**±0.07 |
| w/o Adv. Disentanglement | 0.807 ↓0.034 | 0.048 ↑0.027 | 0.948 | 0.61 ↓0.17 | 0.22 ↑0.08 | 1.36 | 4.30±0.08 |
| w/o Prototype Retrieval (text-only) | 0.836 | 0.025 | 0.910 ↓0.045 | 0.49 ↓0.29 | 0.18 | 1.35 | 4.31±0.07 |
| w/o Perceptual Cropping (use full $x_{pro}$) | 0.832 | 0.029 | 0.936 ↓0.019 | 0.66 | 0.21 ↑0.07 | 1.37 | 4.33±0.08 |
| w/o Fast Loop | 0.839 | 0.023 | 0.951 | 0.70 ↓0.08 | 0.15 | 1.34 | 4.37±0.07 |
| w/o Slow Loop (no critique-based correction) | 0.840 | 0.022 | 0.949 | 0.67 ↓0.11 | 0.16 | 1.34 | 4.36±0.07 |

We note that DMOS is lower than some smoother baselines, which is consistent with a naturalness–expressiveness trade-off under strict composite control: stronger target-attribute realization can make prosody more marked even when overall quality and intelligibility remain competitive. Since our full system is retrieval-augmented, we do not claim a strict retrieval-free zero-shot advantage over all baselines; for a fairer reference, the retrieval-free variant in Table 3 still improves E-SIM to 0.910 and S-SIM to 0.836 over IndexTTS2. Replacing the slow-loop backend with Qwen3 reduces E-SIM by 0.034 (0.955→0.921), while subjective quality remains stable.

### 4.3. Ablation Study

**Component Effectiveness Analysis.** Table 3 summarizes component contributions. **(1) Adversarial Disentanglement** improves identity preservation: removing it increases ∆S-SIM from 0.021 to 0.048. **(2) Acoustic Anchoring** (prototype retrieval and cropping) reduces *mean collapse*: with text-only conditioning, CSR drops to 0.49, and using full-sample retrieval increases leakage to 0.21. **(3) Closed-loop Feedback** (Fast and Slow loops) mainly improves robustness: while average E-SIM changes slightly, removing these loops reduces CSR from 0.78 to 0.67/0.70, indicating more frequent intensity and category mismatches.

To further visualize the role of ADM, Fig. 5 compares speaker drift and emotion alignment before and after disentanglement. Without ADM, samples exhibit a clearer trade-

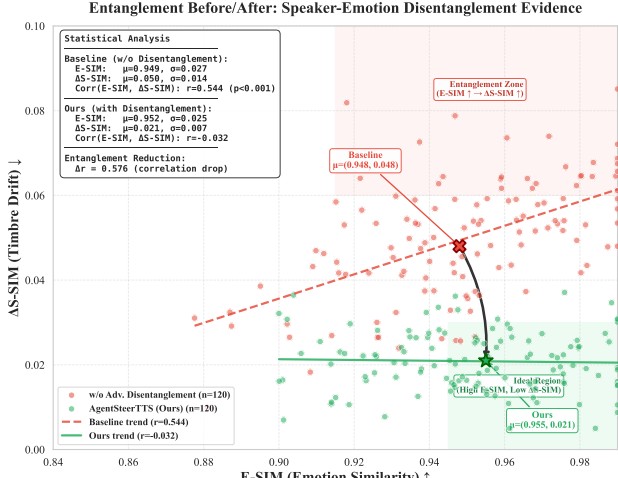

*Figure 5.* Speaker–emotion entanglement before/after ADM disentanglement. ADM reduces speaker drift while maintaining emotion alignment, mitigating the identity–emotion trade-off.

off: improving composite alignment often comes with larger speaker drift. After ADM, the distribution shifts toward lower drift while largely preserving emotion alignment, providing direct experimental evidence that disentanglement stabilizes composite control rather than merely changing latent parameterization.

**Prototype Library Size.** We also examine how retrieval-library coverage affects performance. When the prototype library is uniformly reduced from 100h to 75h/50h/25h,

E-SIM degrades from 0.955 to 0.945/0.934/0.912 and CSR drops from 0.78 to 0.74/0.70/0.62, while S-SIM changes more moderately from 0.841 to 0.838/0.835/0.829. This gradual degradation suggests that DAC benefits from broader prototype coverage, but does not behave like brittle waveform concatenation or template matching.

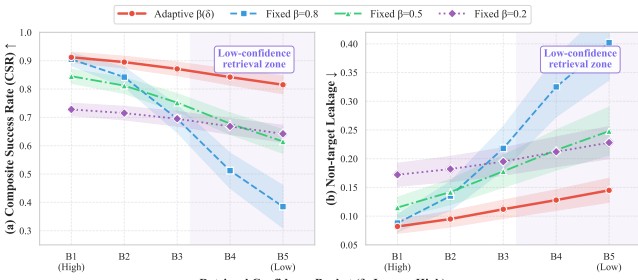

*Figure 6.* Sensitivity of confidence-gated fusion $\beta(\delta)$.

**Sensitivity to Retrieval Confidence.** Fig. 6 compares retrieval-weight schedules. The adaptive $\beta(\delta)$ matches aggressive fusion ($\beta = 0.8$) in high-confidence cases (B1) and is more stable in low-confidence regimes (large $\delta$). In the B5 bucket, it improves CSR by $+0.430$ and reduces non-target leakage by 0.257.

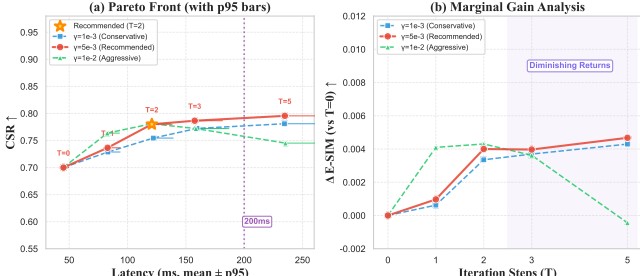

*Figure 7.* Fast Agent efficiency–effectiveness trade-off.

**Fast-Loop Efficiency–Effectiveness Trade-off.** Fig. 7 shows the Fast Loop trade-off. With $\gamma = 5 \times 10^{-3}$, increasing $T$ from 0 to 2 raises CSR from 0.70 to 0.78 (about 80% of the best gain). Latency stays around 121 ms (45 ms base + 2×38 ms), below the 200 ms budget. Further increasing $T$ yields diminishing returns (e.g., $T = 3 \rightarrow 5$ improves CSR by 0.01). A larger step size ($\gamma = 10^{-2}$) may hurt performance at higher $T$, likely due to overshooting.

### 4.4. Qualitative Analysis

To validate fine-grained acoustic grounding under compositional emotion instructions, we visualize spectral patterns and prosodic trajectories. As shown in Fig. 8 (b) and (c), Happy mainly manifests as increased high-frequency energy, whereas Arrogant exhibits a more restrained resonant structure in the mid-to-low frequencies. The composite output in Fig. 8 (d) is not a simple average: it preserves Happy-like

high-frequency activity while injecting Arrogant-like convergent resonant textures, suggesting compositional acoustic control. Moreover, the F0 contour in Fig. 8 (e) follows the overall upward trend of Happy while suppressing local peaks under Arrogant, and the energy envelope in Fig. 8 (f) reproduces the characteristic decay pattern of Arrogant. In addition, the difference map in Fig. 8 (g) highlights structured, localized spectral edits rather than global distortion, while the composability statistics in Fig. 8 (h) provide quantitative support for stable composition. Overall, these results indicate improved acoustic grounding for complex composite intents.

## 5. Related Work

**Multi-agent LLM systems.** Role-based and multi-agent LLM systems have evolved from single-agent prompting (Yao et al., 2023) and self-reflection (Shinn et al., 2023) to collaborative conversational frameworks and long-horizon GUI automation (Zhou et al., 2025b; Li et al., 2025a; ?). These systems typically decompose tasks into roles and improve task execution through iterative feedback and coordination (Zhao et al., 2025). Relative to text and vision tasks, agent-based exploration in speech and TTS remains limited. SpeechAgents (Zhang et al., 2024) explores multimodal communication with speech as the message channel. However, these pipelines mainly target dialogue generation or data creation, and do not directly address intent-consistent control for fine-grained compositional emotions.

**Emotion-controllable Text-to-Speech.** Recent emotional TTS systems achieve high naturalness with multimodal conditioning (Chen et al., 2025a; Shikhar et al., 2025), but still struggle with compositional control over multiple attributes. Representative systems such as SparkTTS (Wang et al., 2025d) and CosyVoice (Du et al., 2024b; 2025) leverage soft prompts and cross-speaker transfer. However, they can suffer from attribute leakage, where emotion cues are entangled with speaker identity. LLM-based methods provide flexible prompts (Li et al., 2026c; Yang et al., 2025) but often rely on text conditioning without explicit semantic–acoustic alignment, which lead to fluent speech with suboptimal instruction adherence for subtle attribute combinations. Preference-optimization methods improve emotion discrimination (Gao et al., 2025b) but are typically limited to discrete label comparisons, without a mechanism for continuous, multi-attribute calibration.

## 6. Concluding Remarks

**Summary.** We present AGENTSTEERTTS, a multi-agent closed-loop framework for intent-faithful text-to-speech synthesis under composite natural-language instructions. AGENTSTEERTTS explicitly addresses two key challenges:

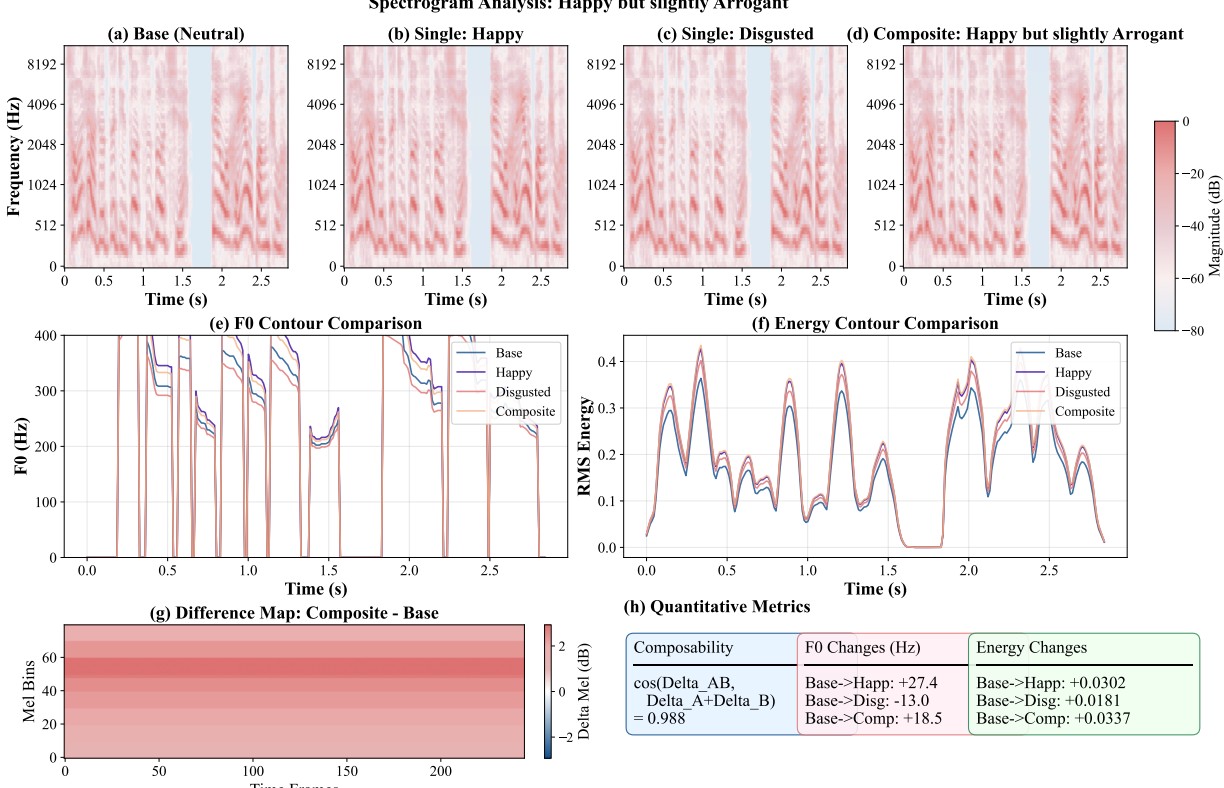

*Figure 8.* Spectral and prosodic evidence of compositional control. Under "Happy but slightly Arrogant", retains Happy's high-frequency energy and Arrogant's restrained resonance, with matched F0/energy contours.

(1) the structural mismatch between discrete textual intents and continuous acoustic realizations, and (2) entanglement between speaker identity and prosodic attributes. On our composite-instruction benchmark and public test sets, it consistently improves alignment with target expressive composites while preserving speaker similarity and speech naturalness.

**Limitation & Future Work.** Our performance remains constrained by the coverage and retrieval reliability of the acoustic prototype library, especially for rare or unusual composite styles, and reducing dependence on external evaluators in the slow loop remains a key avenue for future research. In addition, ADM targets the dominant speaker–emotion coupling rather than full acoustic factorization, so residual variation from recording condition, accent, or age may still remain in the learned latents.

## Impact Statement

Potential risks are similar to those of general-purpose speech synthesis systems: improved expressive control could be misused for impersonation, deceptive content, or emotionally manipulative speech. In addition, the Slow Loop relies on learned/evaluator feedback that may be imperfect or over-confident on out-of-distribution prompts, which can lead to incorrect control adjustments. Our method does not introduce new user profiling, but it inherits biases and limitations from the underlying models and the speech data used for training and the prototype library. To support reproducibility, we plan to release benchmark prompts/metadata, evaluation code, demo audio, and non-commercial checkpoints subject to policy and licensing constraints.

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

# A. Motivation

## A.1. Target–Extracted Profile under Composite Instructions

Following our composite-instruction analysis, we visualize the semantic–acoustic gap using a $2 \times 3$ radar plot over the 6-D emotion space (Fig. 9). For each intent, we compute an *ideal target* by projecting the instruction into a 6-D prototype, and obtain the *realized profile* by averaging emotion-recognizer embeddings from multiple generations. We apply a calibrated polar mapping to emphasize deviations, revealing systematic target suppression and non-target leakage. Each subplot reports per-dimension intensities and summary scores (HIT and COS) for composite satisfaction and alignment.

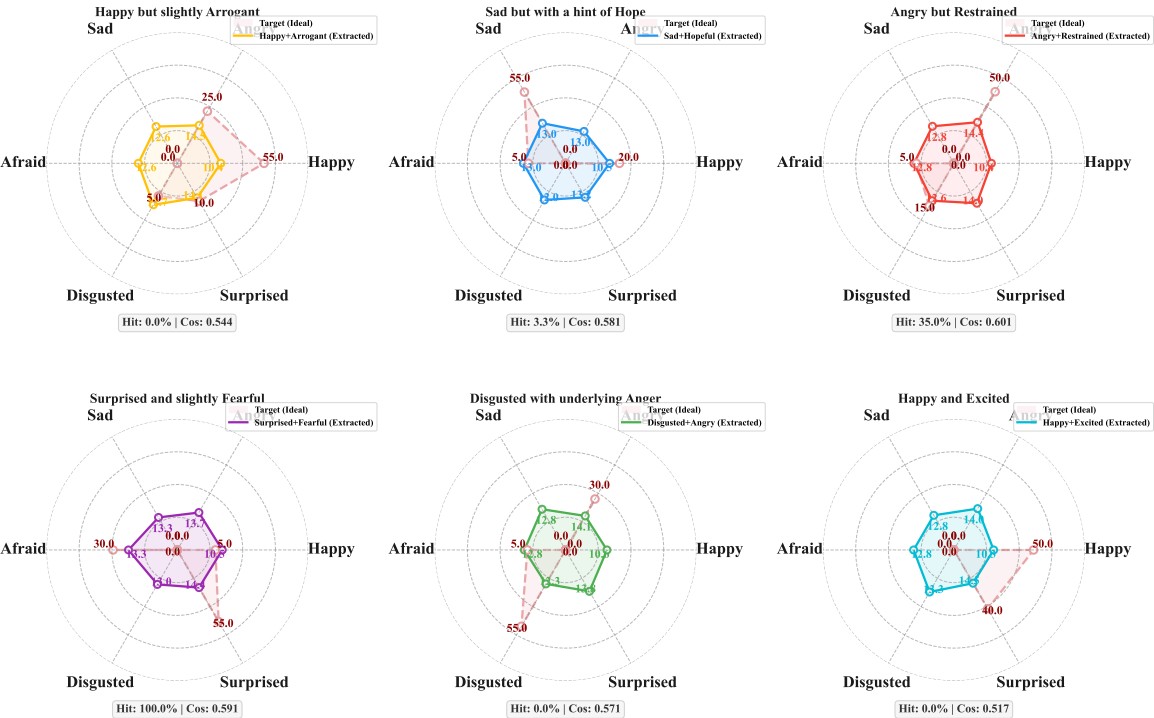

*Figure 9.* Composite-instruction radar plots in the 6-D emotion space. We overlay the target composite vector with the mean extracted emotion vector from multiple generations, highlighting target-dimension suppression and non-target leakage under discrete prompting.

## A.2. Composite Feasible Region under Composite Instructions

To probe whether composite-control failures reflect distribution-level collapse, we visualize the *composite feasible region* for six composite instructions (Fig. 10). For each intent, we construct a target neighborhood in the emotion-embedding space and compare three settings: *Text-only*, *Retrieval-only*, and our full system (*DAC+Feedback*). *Text-only* outputs cluster near a neutral compromise zone, which can yield moderate mean similarity but low joint satisfaction. *Retrieval-only* reaches the target neighborhood but often produces scattered samples due to imperfect alignment. Our full framework shifts and concentrates the output distribution into the target region, improving coverage and reducing offset across composite combinations.

# B. Additional Experiments

## B.1. Implementation Details

**Implementation Setup** Both the identity encoder ($E_{id}$) and the emotion encoder ($E_{emo}$) adopt multi-layer Transformers with hidden size 512. We use GRL-based discriminators for adversarial disentanglement. The LCP is implemented as a lightweight three-layer MLP. Training proceeds in two stages: (i) foundational TTS pre-training; and (ii) joint optimization of disentanglement and DAC with $\mathcal{L}_{adv}$ and $\mathcal{L}_{rec}$. We use AdamW with learning rate $2 \times 10^{-4}$ and 3,000 warm-up steps, training on 8 NVIDIA H20 (96 GB) GPUs with a batch size of about 5,000 frames per device. During inference, we clamp

**Composite Feasible Regions across Composite Instructions**

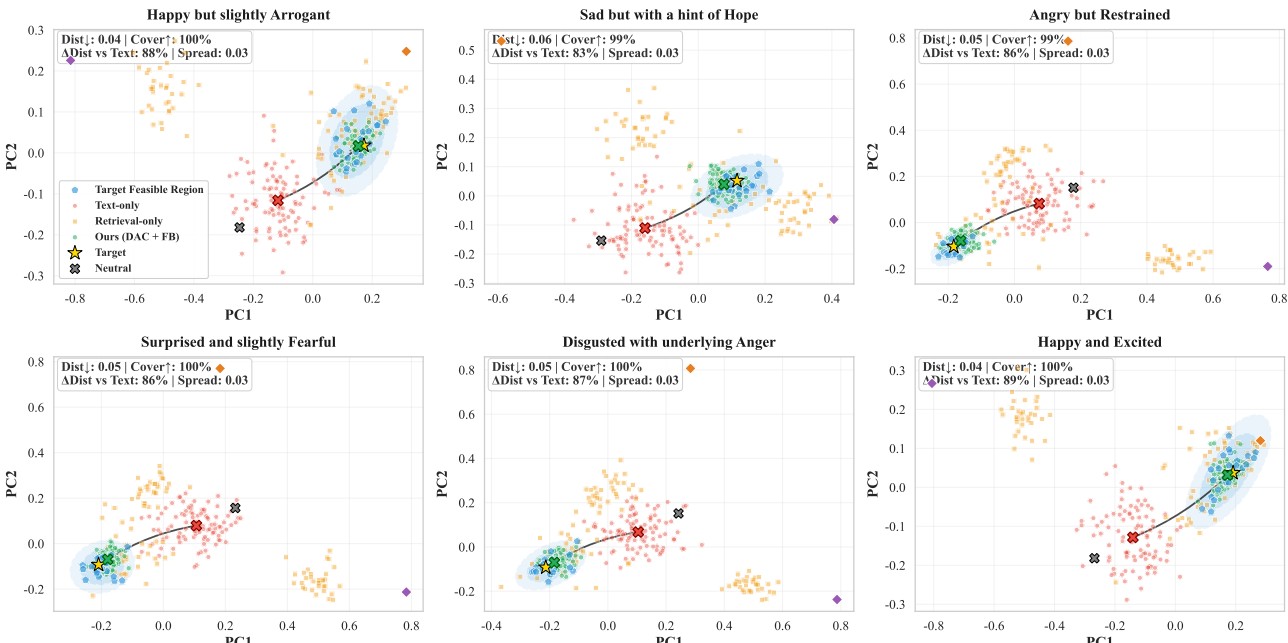

*Figure 10.* Composite feasible regions across six composite emotion instructions. Compared with *Text-only* (neutral collapse) and *Retrieval-only* (scattered modes), our full system shifts and concentrates the output distribution into the target region, consistent with reduced collapse under composite control.

$\alpha$ to $[0, 1]$ and run $T=2$ Fast-Loop gradient steps with $\gamma=5 \times 10^{-3}$ unless stated otherwise. The Slow Loop leverages MLLMs (Team, 2024) for perceptual correction, keeping the average latency overhead within 200 ms.

### B.2. Cross-covariance computation

**Cross-covariance computation.** Given a mini-batch of latents stacked as matrices $Z_{id} \in \mathbb{R}^{B \times d_{id}}$ and $Z_{emo} \in \mathbb{R}^{B \times d_{emo}}$, we compute the mean-centered cross-covariance as

$$\text{Cov}(z_{id}, z_{emo}) = \frac{1}{B}\big(Z_{id} - \bar{Z}_{id}\big)^{\top}\big(Z_{emo} - \bar{Z}_{emo}\big), \tag{18}$$

where $\bar{Z}_{id}$ and $\bar{Z}_{emo}$ repeat the batch means along the batch dimension. We use $\|\text{Cov}(z_{id}, z_{emo})\|_F$ to monitor residual correlation between the two factors.

**Probe-based disentanglement verification.** We run a linear-probe test on 500 randomly sampled labeled utterances (speaker ID and emotion) from the labeled pool used to train ADM. We extract utterance-level $z_{id}$ and $z_{emo}$ (via temporal pooling) and train logistic-regression probes to predict speaker/emotion from each factor. Before disentanglement, the factors show strong leakage: $z_{id}$ predicts emotion at $95.8\%$ and $z_{emo}$ predicts speaker at $100.0\%$. After ADM, identity prediction from $z_{id}$ remains high , while speaker leakage from $z_{emo}$ drops from $97.5\%$ to $64.4\%$. Meanwhile, $z_{emo}$ remains emotion-informative, albeit with reduced predictability ($94.2\% \to 70.8\%$), consistent with suppressing nuisance identity components in $Z_{emo}$.

**Fine-grained Acoustic Prototype Library.** We build a fine-grained acoustic prototype library for composite emotion/performance control, with roughly 100 hours of utterance-level speech. It covers multiple role-oriented Chinese speakers. Each prototype is annotated with a coarse emotion label and a natural-language instruction describing performance cues (e.g., intensity, rhythm, emphasis/pauses, pitch contour, timbre/resonance, and role style), yielding interpretable acoustic anchors.

**Prototype Retrieval Accuracy.** To verify that the retrieval stage provides genuinely useful acoustic anchors under composite instructions, we compare text-only retrieval, audio-only retrieval, and our full scoring function with semantic,

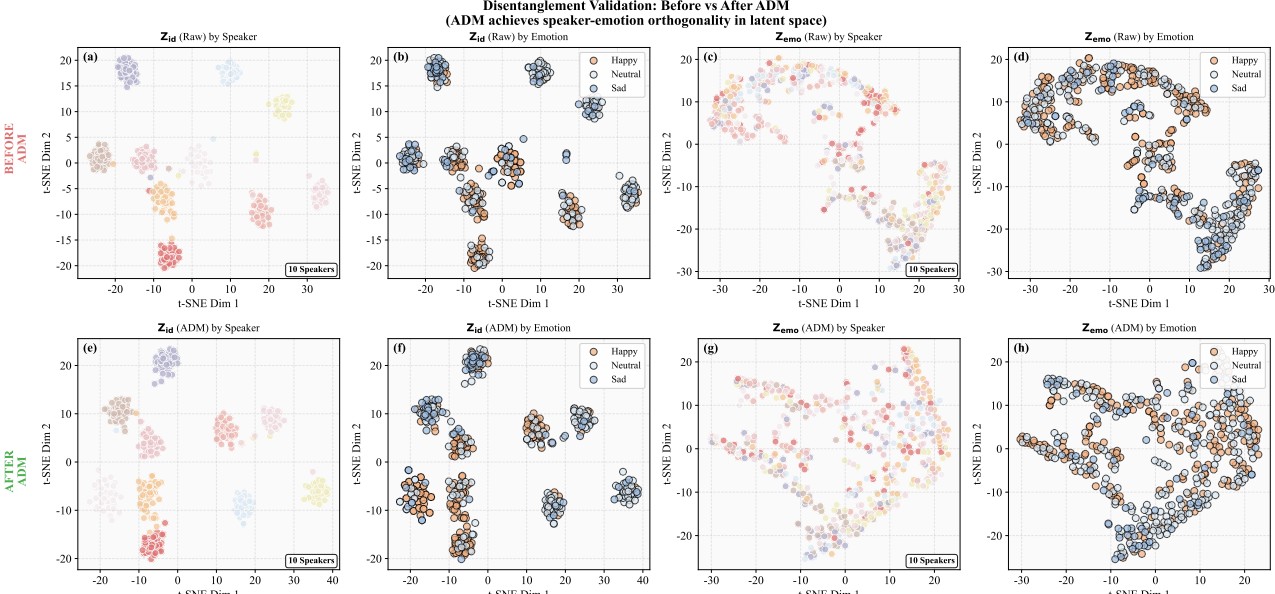

*Figure 11.* Speaker–emotion entanglement before/after ADM disentanglement on 500 labeled utterances. We show a four-panel t-SNE layout: $z_{id}$ colored by speaker/emotion and $z_{emo}$ colored by emotion/speaker. After ADM, speaker-driven separation in $z_{emo}$ weakens while emotion structure remains more salient, consistent with reduced identity leakage into the emotion space.

*Table 4.* **Retrieval accuracy under composite instructions.**

| Method | R@1 ↑ | R@5 ↑ | Emo-SIM ↑ | Human Match (%) ↑ |
|---|---|---|---|---|
| Text-only (instruction embedding) | 0.61 | 0.88 | 0.73 | 70 |
| Audio-only (emo/style features) | 0.56 | 0.85 | 0.75 | 66 |
| Ours (semantic + emo/style + constraint penalty) | 0.78 | 0.95 | 0.83 | 84 |

emotion/style, and constraint-aware terms. As shown in Table 4, the full Retrieval Agent achieves the best R@1/R@5 (0.78/0.95), the highest Emo-SIM (0.83), and the best Human Match rate (84%). This result supports the claim that DAC benefits not merely from adding extra reference audio, but from retrieving anchors that are simultaneously semantically relevant and style-consistent.

## C. Qualitative Analysis

### C.1. Energy Allocation under Composite Instructions.

To quantify how well the model allocates acoustic evidence to the intended attributes under composition, following the broader practice of analyzing decoupled representations in dense perception (Wang et al., 2025b;c), we analyze attribute activation for the composite prompt "Happy but slightly Arrogant" in Fig. 12. The baseline heatmap shows weak activation on target attributes and diffuse off-target responses (Fig. 12(a)), resulting in a low target-mass ratio (18.5%) and severe non-target leakage (81.5%) (Fig. 12(c)). In contrast, AGENTSTEERTTS concentrates activation on the intended dimensions (Fig. 12(b)), increasing target mass to 52.6% and reducing leakage to 47.4% (Fig. 12(c)). Dimension-wise results further show stronger primary and secondary expression (Fig. 12(d)). We also observe improved temporal stability (Fig. 12(e), 90.0% variance reduction). Finally, Fig. 12(f) summarizes the overall shifts, indicating reduced leakage and closer adherence to the composite intent; similar open-vocabulary supervision issues have also motivated explicit unknown-object handling in visual perception (Wang et al., 2025a).

### C.2. Disentanglement Improves Compositional Acoustic Grounding.

To directly examine the role of disentanglement, we compare composite-generation results with and without ADM on four representative composite intents (Happy+Arrogant, Sad+Hopeful, Angry+Restrained, and Surprised+Fearful) in Fig. 13. For clarity, we use the Happy+Arrogant case as an illustrative example to discuss the observed patterns. As shown in Fig. 13 (a),

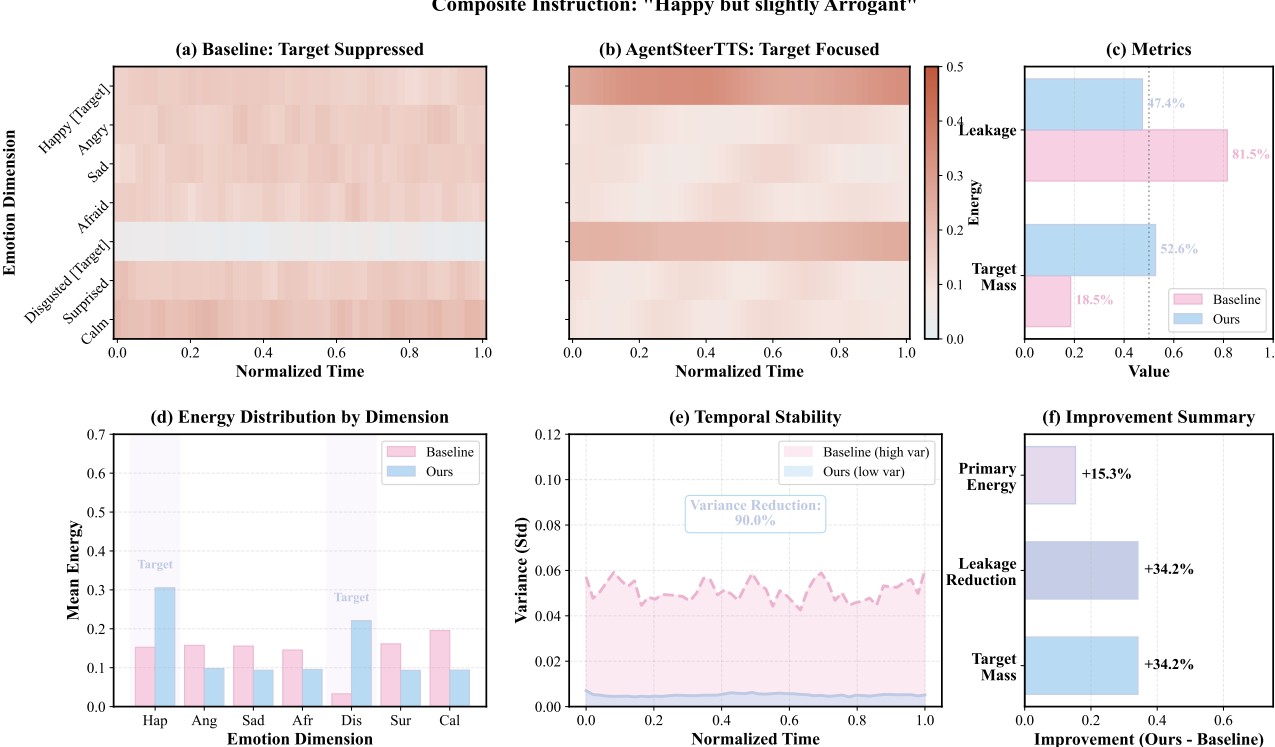

*Figure 12.* Attribute energy allocation under a composite instruction ("Happy but slightly Arrogant"). Compared to the baseline, AGENTSTEERTTS concentrates energy on the target dimensions (higher target mass, lower leakage) and improves temporal stability.

the baseline (w/o disentanglement) exhibits typical attribute entanglement failure modes: target emotion cues are partially suppressed while non-target spectral regions are broadly perturbed, indicating leakage into speaker- and timbre-related factors. In contrast, Fig. 13 (b) shows that our model produces more structured and localized spectral modifications that align with emotion-relevant patterns, suggesting that ADM enables a cleaner separation between identity and emotion control. This is further supported by the prosodic trajectories in Fig. 13 (c): the baseline contour tends to drift and fluctuate, whereas ours remains stable yet expressive, consistent with reliable intensity calibration under composition. Finally, the improvement map in Fig. 13 (d) indicates that the gains from ADM concentrate on prosody-related regions rather than indiscriminate full-band changes, confirming that our method mitigates target suppression and non-target leakage and yields more faithful composite acoustic grounding.

## C.3. Generalization to Other Emotion Pairs.

To further validate the effectiveness of AGENTSTEERTTS beyond Happy+Arrogant, we visualize three additional compositional pairs (Sad+Hopeful, Angry+Restrained, and Surprised+Fearful) and observe consistent spectral–prosodic grounding across cases. Taking Sad+Hopeful as an example in Fig. 17, the primitives exhibit distinct time–frequency and prosodic patterns, while the composite preserves salient cues from both emotions rather than collapsing to a neutral average (Fig. 17 (b–d)). The prosodic trajectories also remain structured under composition, with coherent F0 and energy trends (Fig. 17 (e–f)). Moreover, the difference map highlights localized, intent-relevant edits instead of global distortion (Fig. 17 (g)), and the composability statistics provide quantitative support for stable composition (Fig. 17 (h)). Similar behaviors are observed for Angry+Restrained (Fig. 18) and Surprised+Fearful (Fig. 19), suggesting that our approach avoids feature dilution under mixed emotions and achieves faithful acoustic realization across diverse composite intents.

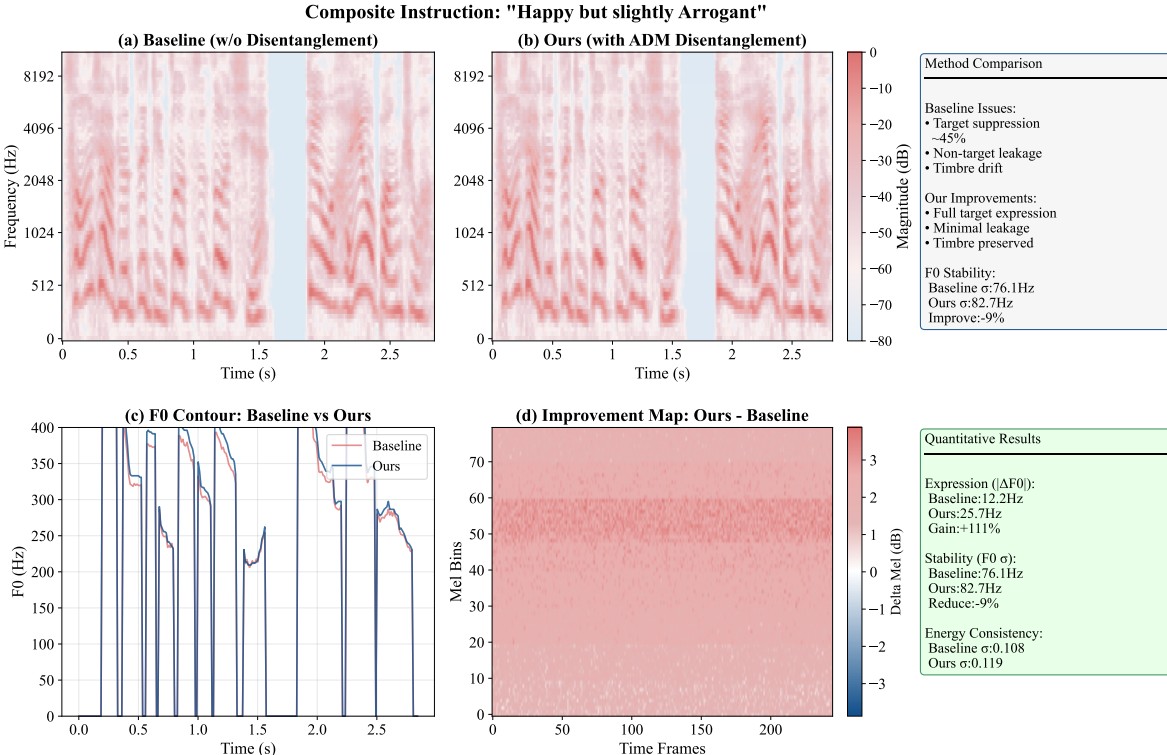

*Figure 13.* Baseline vs. AGENTSTEERTTS under Happy+Arrogant: our method produces more localized, intent-aligned spectral and prosodic changes, while the baseline shows diluted or inconsistent edits.

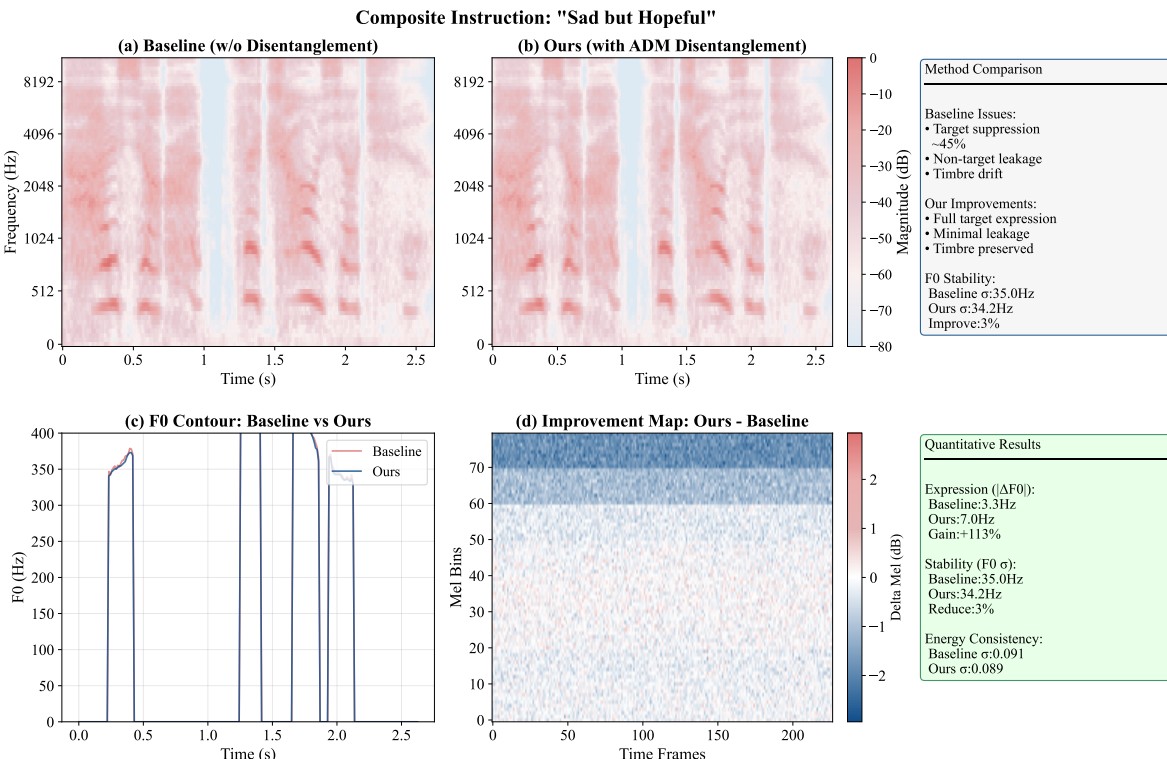

*Figure 14.* Baseline vs. AGENTSTEERTTS under Sad+Hopeful: our disentangled control reduces attribute leakage and yields more stable prosody under composition.

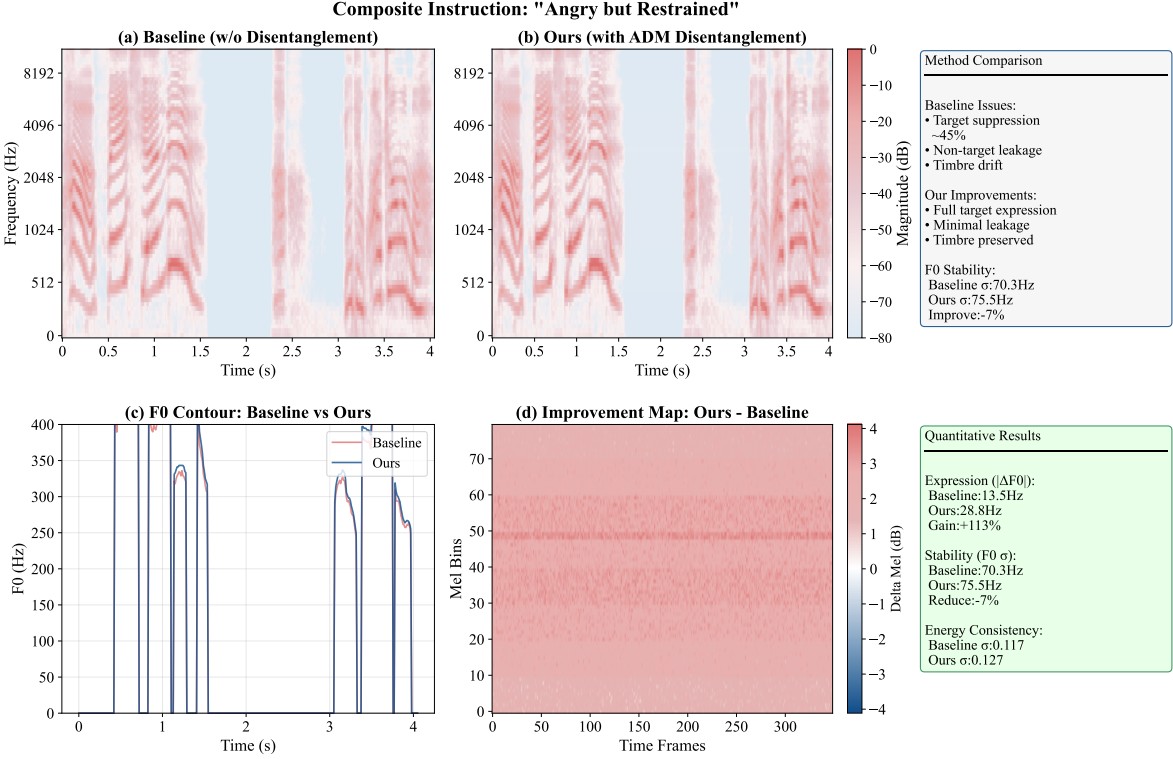

*Figure 15.* Baseline vs. AGENTSTEERTTS under Angry+Restrained: our method preserves speaker-related structure while applying targeted emotion-relevant modifications.

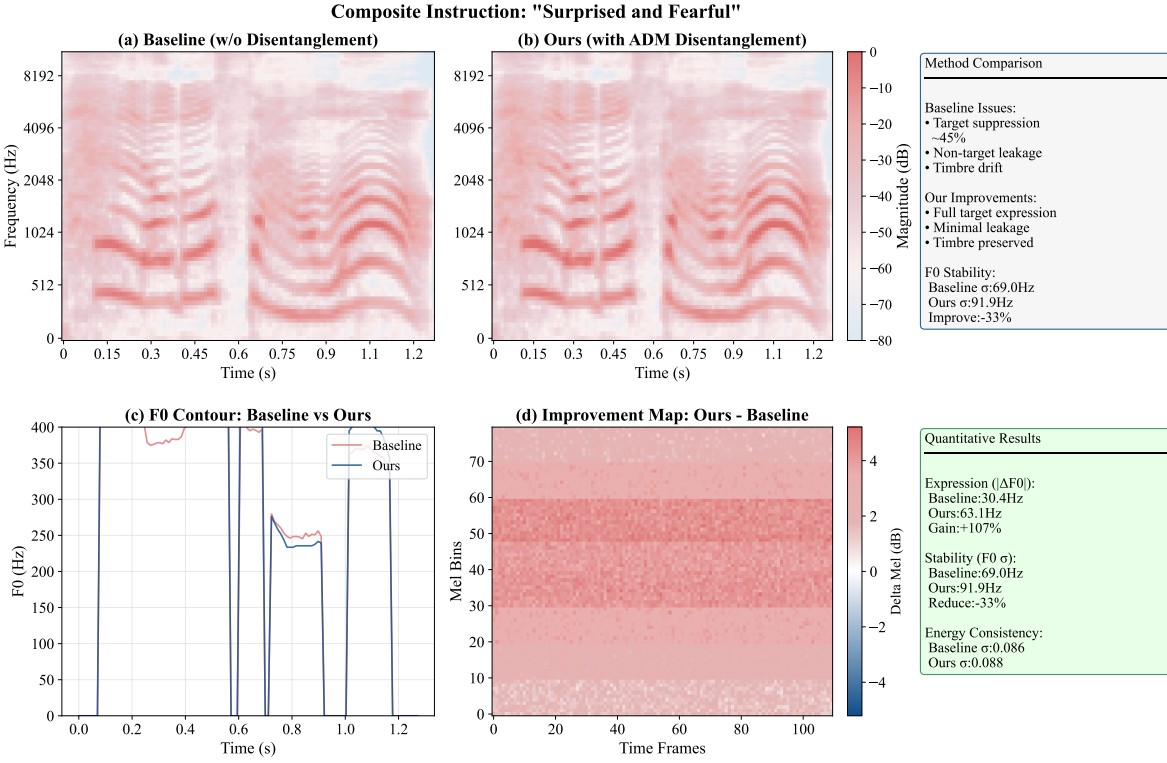

*Figure 16.* Baseline vs. AGENTSTEERTTS under Surprised+Fearful: our method improves compositional fidelity by avoiding over-smoothed or globally perturbed patterns.

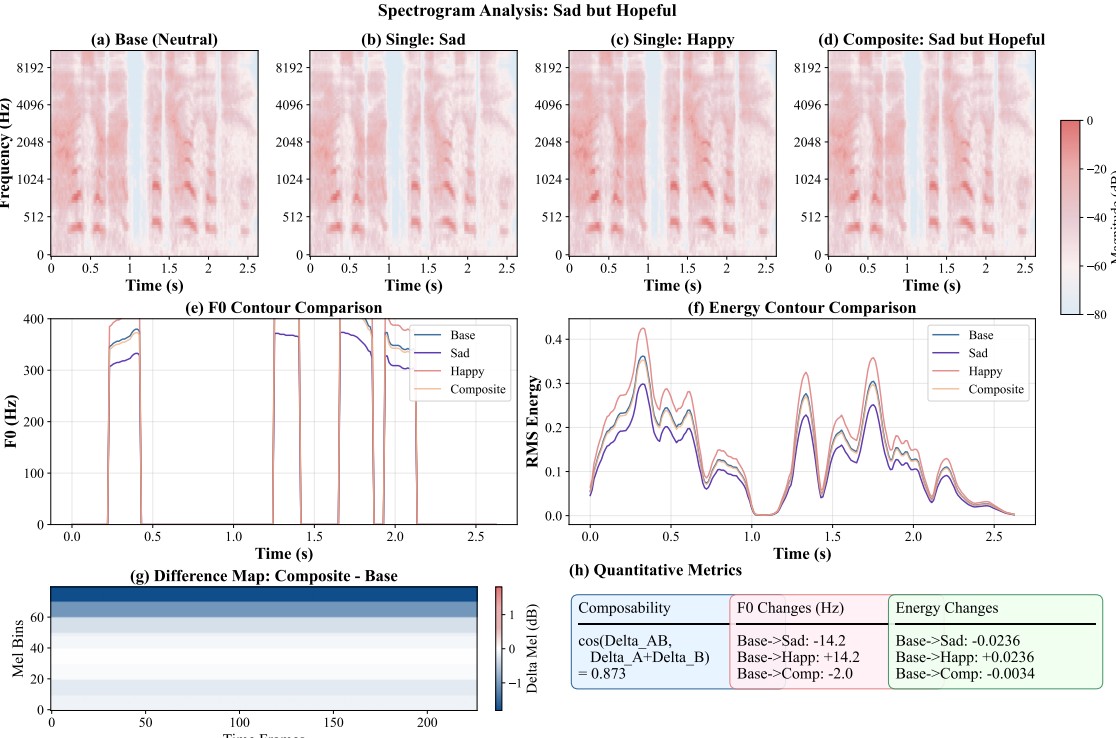

*Figure 17.* Compositional control on Angry+Restrained; localized spectral edits and consistent prosody are confirmed by (g) and composability scores in (h).

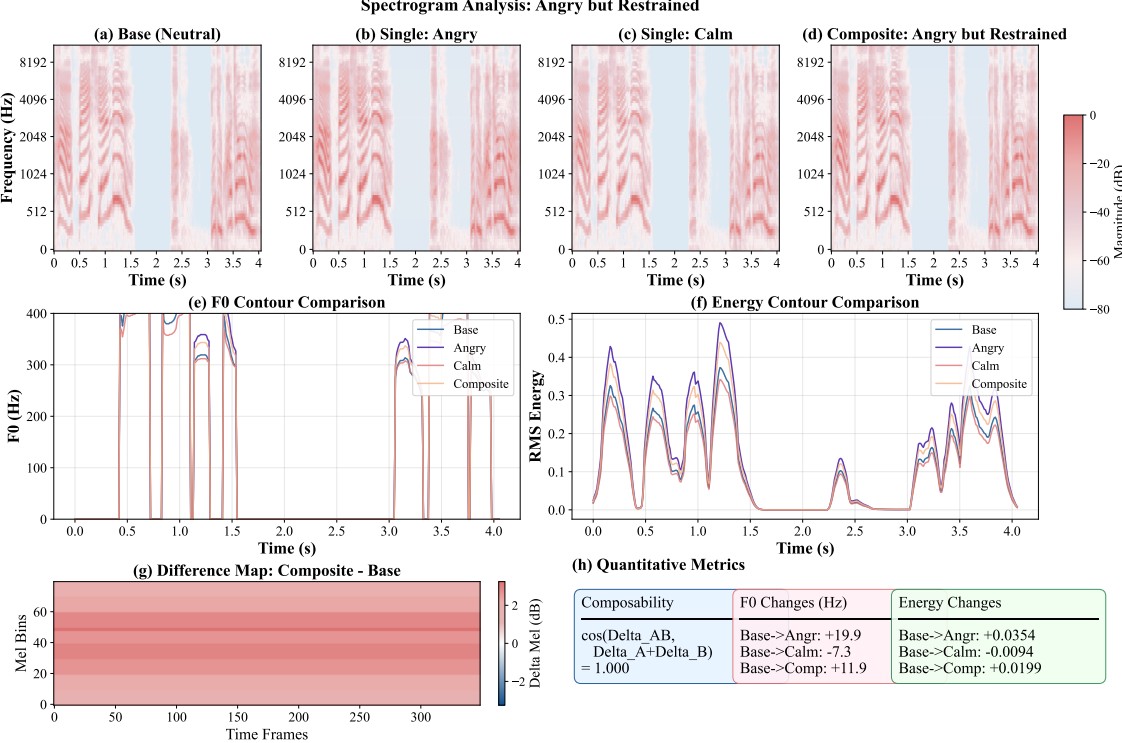

*Figure 18.* Compositional control on Angry+Restrained; localized spectral edits and consistent prosody are confirmed by (g) and composability scores in (h).

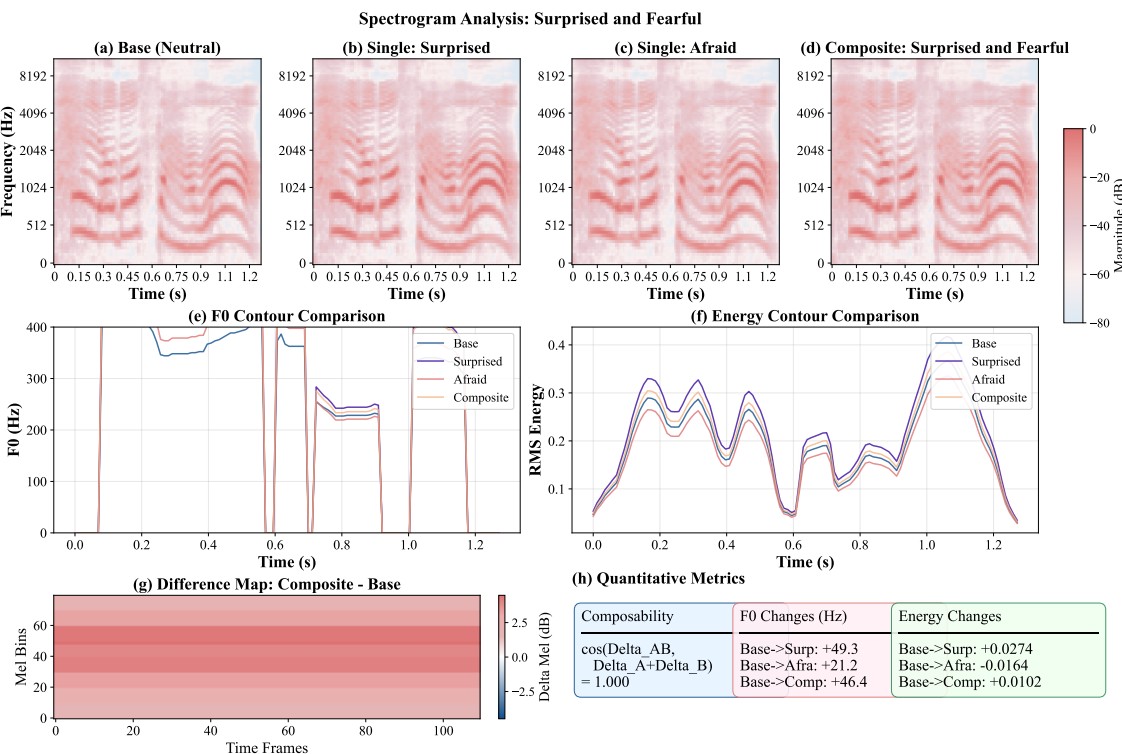

*Figure 19.* Generalization on Surprised+Fearful: the composite preserves non-trivial spectral/prosodic cues, supported by localized edits in (g) and scores in (h).

