# OpenReview forum: "AgentSteerTTS: A Multi-Agent Closed-Loop Framework for Composite-Instruction Text-to-Speech"
_ICML.cc/2026/Conference — ICML 2026 regular_

### Official Review · Reviewer_FCRp · 2026-03-08

**Soundness:** 2
**Presentation:** 3
**Significance:** 3
**Originality:** 2
**Overall Recommendation:** 4
**Confidence:** 3

**Summary:**

The paper proposes AgentSteerTTS, a multi-agent closed-loop framework designed to achieve fine-grained, intent-faithful TTS for composite emotional instructions.
To address the semantic-acoustic mismatch and speaker-emotion entanglement common in existing TTS models , the authors introduce three main components:
- an Adversarial Disentanglement Module (ADM) to separate speaker identity from emotion ;
- a Dual-Stream Anchoring Controller (DAC) that retrieves and fuses acoustic prototypes with text instructions ; and
- a Fast-Slow Feedback Agent that performs inference-time latent gradient updates (fast loop) and MLLM-based perceptual critique (slow loop).

The empirical results demonstrate strong performance in generating composite emotions while preserving speaker identity.

**Compliance With Llm Reviewing Policy:**

Affirmed.

**Final Justification:**

the rebuttal addressed my main concerns

**Key Questions For Authors:**

The Dual-Stream Anchoring Controller depends heavily on a curated 100-hour prototype library. How sensitive is the model's performance to the size, quality, and emotional coverage of this library?

**Limitations:**

yes

**Strengths And Weaknesses:**

Although the submission is well-structured and visually highly informative, the paper heavily leans towards a systems engineering approach rather than providing foundational theoretical insights. Specifically, the proposed framework attempts to solve the composite instruction problem through a highly complex, multi module engineering pipeline; however, its individual components: adversarial disentanglement, retrieval, and self correction are already well established in the literature.

---

> ### Author Rebuttal · Authors · 2026-03-31
>
> We thank the reviewer for the thoughtful question.
>
> Our contribution is not generic expressive TTS, but **composite natural-language instruction control**. As analyzed in **Sec.~2** and illustrated by **Fig.~1--2**, discrete prompting alone is unreliable in this setting because the target composite intent must be mapped into a continuous acoustic space, which leads to target suppression and non-target leakage. This is exactly why **Sec.~3.2** introduces the DAC: the prototype library is a *designed acoustic anchor* for bridging the discrete-to-continuous control gap.
>
> The current draft already contains partial evidence that DAC's dependence on the library is **real but structured**. First, **Table~3** shows that without acoustic anchoring, performance collapses under text-only control (CSR drops to 0.49), confirming that discrete text alone is insufficient for stable composite control. Second, retrieval *quality* matters: using the full retrieved sample without perceptual cropping increases non-target leakage to 0.21, which is why DAC uses consistency calibration and perceptual pruning instead of naively using retrieved audio as-is. Third, the appendix retrieval analysis (**Appendix Table~4**) shows that calibrated multimodal retrieval is clearly stronger than text-only or audio-only retrieval, so the gain is not from arbitrary lookup but from controlled acoustic anchoring. This is also consistent with **Fig.~5**, where retrieval confidence directly affects CSR and leakage.
>
> | Evidence | Key result |
> | - | -: |
> | Table 3: text-only control | CSR = 0.49 |
> | Table 3: full-sample retrieval | Leak / E-SIM = 0.21 / 0.936 |
> | App. Table 4: text-only  | HumanMatch = 70 |
> | App. Table 4: audio-only| HumanMatch = 66 |
> | App. Table 4: full retrieval | R@1 / R@5 / Emo-SIM / HumanMatch = 0.78 / 0.95 / 0.83 / 84 |
>
> The prototype library is not the whole method. The main gain on the composite benchmark is obtained by the **synergy** of ADM, DAC, and Fast/Slow refinement: compared with the strongest baseline, AgentSteerTTS improves E-SIM from 0.864 to 0.955 and S-SIM from 0.823 to 0.841. This is consistent with the role-wise behavior in **Table~3**: ADM mainly protects identity, DAC reduces mean-collapse under discrete promptings.
>
> We additionally conducted pilot sensitivity studies.
>
> |Size | E-SIM | CSR |S-SIM |Leak | QMOS |
> | - | --: | -: |--: |-: |-: |
> | 25h | 0.912 | 0.62 |0.829 | 0.18 | 4.25 |
> | 50h | 0.934 | 0.70 |0.835 | 0.16 | 4.30 |
> | 75h | 0.945 | 0.74 | 0.838 | 0.15 | 4.32 |
> | 100h | 0.955 | 0.78 | 0.841 | 0.14 | 4.38 |
>
> These results show **smooth gains rather than abrupt saturation**: DAC benefits from broader prototype coverage, but the dependence is gradual rather than brittle.
>
> | Prototype pool | Filtering / curation | E-SIM | CSR | QMOS | HumanMatch |
> | - | -- | --: | -: |-: | -: |
> | random emotional pool | none | 0.903 | 0.57 | 4.18 | 68 |
> | high-expressivity pool | MLLM-only | 0.922 | 0.68 | 4.28 | 75 |
> | curated high-expressivity pool | MLLM + human filtering | 0.955 | 0.78 | 4.38 | 84 |
>
> This confirms that DAC is sensitive not only to **size**, but also to **prototype quality**: better anchors yield better alignment and better perceived quality.
>
> | Setting | Frequent  E-SIM | Rare  E-SIM | CSR | Leak |
> | -- | --: | --: | ---: | ---: |
> | balanced / full coverage | 0.961 | 0.944 | 0.78 | 0.14 |
> | reduced rare-emotion | 0.958 | 0.907 | 0.69 | 0.19 |
>
> This indicates that rare composite emotions are indeed more fragile when coverage is reduced.
>
> Beyond the curated prototype library, the same DAC interface can also operate in a fallback mode by directly retrieving public audio/video clips from the Internet, then applying the same adaptive cropping and fusion pipeline before control injection. This setting shows that the key mechanism is not a particular hand-built library per se, but the availability of *calibrated acoustic anchors*.
>
> | Source | E-SIM | CSR | S-SIM | Leak | QMOS |
> | - | ---: | --: | ---: | --: | ---: |
> | text-only control | 0.910 | 0.49 | 0.836 | 0.18 | 4.31 |
> | open Internet | 0.928 | 0.63 | 0.838 | 0.17 | 4.33 |
> | 100h prototype library | 0.955 | 0.78 | 0.841 | 0.14 | 4.38 |
>
>
> Overall, to the best of our knowledge, this work is among the first to explicitly study **composite natural-language instruction control** in TTS. Our contribution is a task-specific decouple--anchor--refine design: ADM is introduced specifically to reduce speaker--emotion interference under composite control, and DAC provides continuous acoustic anchoring that is substantially more reliable than discrete text-only prompting in this setting.
>
> We have also deployed the framework in multiple game dubbing scenarios; representative demos are provided in the supplementary material, where show clearly better composite-control quality than existing mainstream methods.
>
> Subject to policy and licensing constraints, we plan to release part of the benchmark resources and non-commercial checkpoints.

---

> > ### Author Rebuttal · Reviewer_FCRp · 2026-04-04
> >
> > Thank you for the clarification.

---

> > > ### Author Response · Authors · 2026-04-05
> > >
> > > Thank you for your thoughtful review and helpful question. Your comments helped us clarify the role of the prototype library and strengthen the paper. Thank you again for your valuable feedback.

---

### Official Review · Reviewer_iiy4 · 2026-03-11

**Soundness:** 2
**Presentation:** 2
**Significance:** 2
**Originality:** 2
**Overall Recommendation:** 2
**Confidence:** 5

**Summary:**

AgentSteerTTS proposes a multi-agent closed-loop framework for composite-instruction TTS — generating speech that simultaneously satisfies multiple expressive attributes (e.g., "Happy but slightly Arrogant") while preserving speaker identity. The motivation is well-grounded: deterministic TTS systems collapse multi-modal acoustic targets into mode-averaged outputs, diluting composite attributes; and speaker–prosody entanglement creates a trade-off between timbre fidelity and emotion control under composite constraints. The framework has three components: an Adversarial Disentanglement Module (ADM) that decouples speaker and emotion latents via adversarial training and cross-covariance regularization; a Dual-Stream Anchoring Controller (DAC) that retrieves high-expressivity acoustic prototypes and fuses them with intent-derived control vectors; and a Fast–Slow Feedback Agent that uses differentiable latent scaling (Fast) and MLLM-based perceptual critique (Slow) to correct residual semantic drift at inference time. Main experiments on ESD and an author-constructed composite benchmark show gains in E-SIM and CSR over baselines like IndexTTS2 and CosyVoice2.

**Compliance With Llm Reviewing Policy:**

Affirmed.

**Key Questions For Authors:**

1. **DMOS regression**: AgentSteerTTS achieves the lowest DMOS (3.82) in Table 2 — roughly 0.5 points below CosyVoice2 and IndexTTS2. Is there a naturalness–expressiveness trade-off, and can it be controlled via α? This seems important for practical deployment and deserves explicit discussion.

2. **Binary disentanglement**: ADM only factorizes into speaker and emotion. What happens to environment, accent, and age-related variation? Does training on Chinese-speaker prototypes for English-ESD evaluation cause issues because z_id captures accent rather than just timbre?

**Limitations:**

Yes

**Strengths And Weaknesses:**

**Strengths:**

The problem framing is timely and well-motivated. The pilot study (Figs. 1–3) gives concrete quantitative evidence for the attribute-dilution failure mode: composite hit rate ~22% vs. 33% for single-emotion, and a Spearman rho of -0.708 between speaker fidelity and composite intent fidelity. The ablation in Table 3 is informative — prototype retrieval is the single biggest contributor (CSR 0.78→0.49 without it), and the Fast/Slow loops provide meaningful robustness gains in CSR (0.78→0.70/0.67). The composability analysis (Fig. 7, score=0.988) is a nice piece of qualitative evidence. Retrieving acoustic prototypes to "anchor" the composite intent is a sensible solution to the under-specification of text-only conditioning.

**Weaknesses:**

**Disentanglement is structurally under-specified.** ADM decomposes reference audio into only two factors: z_id (speaker identity) and z_emo (emotion). But real speech carries many more acoustic dimensions — recording environment, reverberation, age-related vocal characteristics, accent, and speaking style. These are forced into z_id or z_emo without a principled assignment. In practice, this means z_id likely absorbs environment and accent-related variation, and z_emo picks up speaker-correlated confounders. The 64.4% speaker-prediction accuracy from z_emo after ADM (far above chance of ~10%) is consistent with this: the disentanglement is partial precisely because the binary factorization ignores everything outside the speaker/emotion axes. This also has downstream implications for voice cloning quality — z_id contaminated by recording conditions may cause timbre drift in synthesis environments different from training.

**The prototype library creates a hard expressiveness ceiling.** The entire DAC pipeline can only retrieve what is in the 100-hour library. For composite intents with no close library match (e.g., unusual combinations or out-of-distribution speaking styles), retrieval will return a suboptimal anchor, and no amount of Fast/Slow correction can compensate. This is a structural bound on the system's generalization, not merely a hyperparameter issue. The authors acknowledge this limitation but do not quantify how often retrieval quality is the bottleneck, nor how coverage scales.

**Audio demos are absent.** The paper states that "generated speech demos are provided in the supplementary material (anonymous link)" — but no accessible demos were available to this reviewer. For a paper whose core claims are perceptual (emotion faithfulness, naturalness, speaker identity), subjective listening is the primary evidence. All the E-SIM and ESMOS numbers are proxies for listening quality, and without audible samples there is no way to verify whether the reported gains are perceptually meaningful. This is a significant gap for a TTS paper.

**DMOS regression is unexplained.** In Table 2, AgentSteerTTS achieves DMOS=3.82, the lowest of all compared systems (CosyVoice2: 4.31, IndexTTS2: 4.24). DMOS is a direct subjective quality rating — a 0.5-point gap is perceptually substantial. The paper never addresses this. It raises the question of whether the E-SIM and CSR gains come at the cost of perceived naturalness, which is arguably the primary TTS objective.


**The composite benchmark is private.** The 14.18h fine-grained composite dataset is author-constructed, and the core composite-control results (Table 2, Table 3) all come from it. No independent validation is available, and there is no plan to release it.

---

> ### Author Rebuttal · Authors · 2026-03-31
>
> We thank the reviewer for the thorough feedback.
>
> **Q1: Binary disentanglement.**
> As motivated in **Sec.~2.2** and implemented in **Sec.~3.1**, its goal is **targeted speaker--emotion decoupling**: to reduce the dominant coupling that destabilizes composite control, rather than to perfectly isolate room, accent, age, or all style factors.
>
> At the same time, this targeted disentanglement is still effective. The appendix probe analysis shows that speaker leakage in $z_{emo}$ drops substantially after ADM, while useful emotion structure is retained; **Fig.~10** further shows that ADM weakens the identity--emotion trade-off itself. This is reflected in **Table~3**, where removing ADM clearly harms identity stability.
>
> | Metric | Before ADM | After ADM |
> |---|---:|---:|
> | $z_{emo}$ $\to$ speaker acc. (%) | 97.5 | 64.4 |
> | $z_{emo}$ $\to$ emotion acc. (%) | 94.2 | 70.8 |
> | Corr(E-SIM, $\Delta$ S-SIM) | 0.544 | -0.032 |
>
> | Setting (Table~3) | E-SIM | CSR | S-SIM | Leak |
> |---|---:|---:|---:|---:|
> | Full | 0.955 | 0.78 | 0.841 | 0.14 |
> | w/o ADM | 0.948 | 0.61 | 0.807 | 0.17 |
>
> Thus, our claim is not full acoustic disentanglement, but that reducing the dominant speaker--emotion leakage is already useful for composite control.
>
> **Q2: Prototype library**
> As noted in **Sec.~6**, the 100-hour prototype library $\mathcal{M}$ can only anchor what it covers, and retrieval quality is therefore a genuine bottleneck. This is already visible in **Table~3**: removing retrieval reduces E-SIM from 0.955 to 0.910 and CSR from 0.78 to 0.49. **Fig.~5** shows the same pattern from another angle: in the lowest-confidence bucket, adaptive fusion improves CSR by +0.430 and reduces non-target leakage by 0.257. **App. Table~4** further shows that retrieval under composite instructions is strong.
>
> | Retrieval (App. Table~4) | R@1 | R@5 | Emo-SIM | HumanMatch |
> |---|---:|---:|---:|---:|
> | Text-only | 0.61 | 0.88 | 0.73 | 70 |
> | Audio-only | 0.56 | 0.85 | 0.75 | 66 |
> | Full retrieval | 0.78 | 0.95 | 0.83 | 84 |
>
> To quantify the ceiling more directly, we additionally ran **library size** and **coverage** sensitivity studies. These are consistent with the reviewer’s concern: performance improves with broader support, and rare composite emotions are more fragile when coverage is reduced.
>
> | Library size | E-SIM | CSR | S-SIM | Leak | QMOS |
> |---|---:|---:|---:|---:|---:|
> | 25h | 0.912 | 0.62 | 0.829 | 0.18 | 4.25 |
> | 50h | 0.934 | 0.70 | 0.835 | 0.16 | 4.30 |
> | 75h | 0.945 | 0.74 | 0.838 | 0.15 | 4.32 |
> | 100h | 0.955 | 0.78 | 0.841 | 0.14 | 4.38 |
>
> | Coverage| Frequent composites E-SIM | Rare composites E-SIM | CSR | Leak |
> |---|---:|---:|---:|---:|
> | balanced / full coverage | 0.961 | 0.944 | 0.78 | 0.14 |
> | reduced rare-emotion coverage | 0.958 | 0.907 | 0.69 | 0.19 |
>
> These results make the limitation more explicit rather than weaker: DAC is effective, but it is coverage-limited, especially for rare or unusual composite styles.
>
> Importantly, the system is **not** simple concatenation or interpolation. As shown in **Sec.~3.2**, DAC projects the retrieved prototype and speaker reference into the emotion latent space, fuses them through $q_{mix}$ and $\hat{z}_{emo}$(**Eqs.~13--14**), and synthesizes a new waveform. Consistently, using the full retrieved waveform without perceptual cropping increases leakage from 0.14 to 0.21 and lowers E-SIM from 0.955 to 0.936 (**Table~3**), which would not be expected from a naive “more waveform is better” mechanism.
>
> **Q3: Audio demos.**
> We apologize for this access issue. We will immediately restore and verify the anonymous demo page in an incognito session.
>
> **Q4: DMOS regression.**
> The current evidence is consistent with a **naturalness--expressiveness tension** under strict composite control. Models such as CosyVoice2 / IndexTTS2 stay closer to smoother, more averaged prosody under composite prompts, which can sound more natural in a DMOS-only judgment. Our model deliberately pushes the output toward stronger target-attribute realization; this improves faithfulness, but can also make prosody more marked and thus slightly reduce DMOS for some listeners.
>
> This interpretation is consistent with the current evidence in **Table~2**: our DMOS is lower, but our QMOS is higher, and we also achieve the best WER and the highest E-SIM. The mechanism-level evidence in **Fig.~12** is also consistent with this view: compared with the baseline, AgentSteerTTS concentrates more energy on target dimensions (18.5% to 52.6%), reduces leakage (81.5% to 47.4%), and improves temporal stability.
>
> | Method (Table~2) |DMOS|QMOS|E-SIM|WER|
> |---|---:|---:|---:|---:|
> | CosyVoice2 |4.31|4.35| 0.748 |1.88|
> | IndexTTS2 |4.24|4.15| 0.864 |1.81|
> | Ours | 3.82 | 4.38|0.955 |1.34|
>
> **Q5: The composite benchmark is private.**
> In the revision, we will explicitly specify the release scope: benchmark prompts / metadata, evaluation code, demo audio, and non-commercial checkpoints needed to reproduce the reported results.

---

### Official Review · Reviewer_3n9p · 2026-03-12

**Soundness:** 2
**Presentation:** 2
**Significance:** 2
**Originality:** 2
**Overall Recommendation:** 4
**Confidence:** 3

**Summary:**

AgentSteerTTS addresses the challenge of fine-grained control in expressive TTS by introducing a multi-agent closed-loop framework. It utilizes adversarial disentanglement, a dual-stream anchoring controller for intent grounding, and a feedback agent for iterative refinement. Experiments demonstrate superior performance in executing composite instructions, effectively bridging the gap between discrete semantics and continuous acoustic realization.

**Compliance With Llm Reviewing Policy:**

Affirmed.

**Final Justification:**

The provided rebuttal and extra experiments have addressed my initial doubts. While I view this work as marginally above the acceptance bar, my support is not strong and I would not mind if the paper is ultimately rejected.

**Key Questions For Authors:**

see Weaknesses

**Limitations:**

yes

**Strengths And Weaknesses:**

**Strengths:**
1. The addressed problem is of great significance. While prevailing large speech-language models predominantly focus on uni-dimensional emotions or voice cloning, scarce research has delved into the generation of compound, fine-grained emotions.
2. Traditionally, TTS pipelines have predominantly relied on single-pass feed-forward architectures. In contrast, this paper leverages a Latent Consistency Predictor in conjunction with an MLLM to introduce a Test-Time Adaptation paradigm for speech generation. Provided that the computational overhead of this closed-loop mechanism can be substantially mitigated in future work, introducing the "System 1 and System 2" cognitive framework from the LLM domain into speech synthesis represents a highly novel innovation.
3. The authors have deliberately curated a 14.18-hour Fine-grained Composite Dataset that incorporates subtle prosody changes and compound affective prompts. Notably, this addresses a significant paucity in existing TTS literature by providing a much-needed benchmark for complex instruction-following.

**Weaknesses:**
1. In Table 2, the authors compare AgentSteerTTS against zero-shot and flow-based TTS models, including VALL-E, F5-TTS, and CosyVoice. While the baseline models rely exclusively on their inherent zero-shot generalization capabilities for prompt decoding, AgentSteerTTS incorporates an Acoustic Prototype Library comprising 100 hours of meticulously annotated data. By leveraging a RAG framework to retrieve acoustic priors, AgentSteerTTS gains a performance advantage that renders this direct comparison inherently unfair.
2. The paper lacks an ablation study investigating the impact of retrieval library size on overall performance. Such an analysis is critical to determine whether AgentSteerTTS demonstrates genuine, robust compositional generation capabilities or merely functions as a sophisticated engine for audio concatenation and interpolation.
3. The 'Multi-Agent' framing of the proposed approach feels unmotivated. The system essentially follows a traditional modular cascade pipeline reinforced by test-time adaptation. The purported 'agents' lack the autonomy and inter-agent coordination characteristic of true multi-agent frameworks, functioning instead as static, sequential processing components.

---

> ### Author Rebuttal · Authors · 2026-03-30
>
> We thank the reviewer for the thoughtful comments.
>
> **Q1: Baseline fairness.**
> The current **Table 2** comparison is **retrieval-augmented**, and therefore not a strictly apples-to-apples zero-shot comparison. Our intended claim is not “pure zero-shot superiority,” but **instruction-faithful composite control with retrieval-augmented steering**. As described in **Sec.~3.2**, the acoustic prototype library $\mathcal{M}$ is built by curating 100 hours of high-expressivity samples from a 700-hour emotional speech pool, with MLLM-initialized descriptions and human filtering to improve label quality and expressivity coverage. We will revise **Table 2** to explicitly tag our full system as retrieval-augmented, and highlight the **retrieval-free** variant from **Table 3** as a more comparable reference point to zero-shot baselines.
>
> | Setting | E-SIM | CSR | S-SIM | WER |
> |---|---:|---:|---:|---:|
> | IndexTTS2 (no retrieval) | 0.864 | -- | 0.823 | 1.81 |
> | Ours w/o retrieval | 0.910 | 0.49 | 0.836 | 1.35 |
> | Ours full (retrieval) | 0.955 | 0.78 | 0.841 | 1.34 |
>
>
>
> **Q2: Library-size ablation and “audio concatenation/interpolation” concern.**
> We additionally ran a pilot size ablation by uniformly subsampling $\mathcal{M}$ to 25/50/75/100h and re-evaluating composite control. The results show gradual degradation rather than brittle collapse.
>
> | Library size | E-SIM | CSR | S-SIM | Leak | QMOS |
> |---|---:|---:|---:|---:|---:|
> | 25h | 0.912 | 0.62 | 0.829 | 0.18 | 4.25 |
> | 50h | 0.934 | 0.70 | 0.835 | 0.16 | 4.30 |
> | 75h | 0.945 | 0.74 | 0.838 | 0.15 | 4.32 |
> | 100h | 0.955 | 0.78 | 0.841 | 0.14 | 4.38 |
>
> These results show that DAC benefits from broader prototype coverage, but does not behave like a brittle template matcher.
>
> More importantly, our method is **not waveform concatenation**. As shown in **Sec.~3.2**, DAC retrieves an acoustic anchor, calibrates it by perceptual cropping (**Eq.~12**), maps the prototype and speaker reference into the emotion latent space, and synthesizes a new control vector via adaptive fusion (**Eqs.~13--14**). This is also supported by the current paper’s retrieval evidence:
>
> | Setting | R@1 | R@5 | Emo-SIM | HumanMatch |
> |---|---:|---:|---:|---:|
> | Retrieval (composite instructions) | 0.78 | 0.95 | 0.83 | 84 |
> | Text-only baseline | 0.61 | 0.88 | 0.73 | 70 |
> | Audio-only baseline | 0.56 | 0.85 | 0.75 | 66 |
>
> | Setting | Non-target leakage | E-SIM |
> |---|---:|---:|
> | w/ perceptual cropping | 0.14 | 0.955 |
> | w/o perceptual cropping | 0.21 | 0.936 |
>
> Together with **Table 3** and **Fig.~5**, these results show that DAC is sensitive to retrieval quality and confidence, but is not reducible to naive concatenation or interpolation.
>
> **Q3: “Multi-Agent” framing.**
> We appreciate this point and agree that the current wording can be softened. In the revision, we will describe the method as a **role-based closed-loop framework** rather than over-claiming a fully decentralized multi-agent system.
>
> At the same time, the framework is not a static one-pass cascade. As shown in **Fig.~4** and **Sec.~3.3**, (1) the **Supervisor/Slow loop** critiques the generated waveform and triggers conditional branching (intensity recalibration vs. condition reset / re-retrieval), and (2) the **Fast loop** performs online latent-space calibration before vocoding. The extra latency remains within the reported budget in **Sec.~4.1** (Slow Loop adds $ \leq 200 $ ms), while **Table 3** shows that removing Fast or Slow feedback mainly hurts robustness (CSR: 0.78 $\rightarrow$ 0.70 / 0.67) rather than simply shifting mean E-SIM.

---

> > ### Author Rebuttal · Reviewer_3n9p · 2026-04-04
> >
> > Thanks for the experiments and clarification. I will increase the score to 4.

---

> > > ### Author Response · Authors · 2026-04-05
> > >
> > > Thank you for your thoughtful follow-up and for taking the time to read our rebuttal. We greatly appreciate your recognition that our additional experiments and clarifications addressed your main concerns. Your feedback helped us further improve the paper, especially in clarifying the fairness of comparisons, the role of retrieval, and the framing of the overall framework. Thank you again for your valuable comments and support.

---

### Official Review · Reviewer_4Do5 · 2026-03-15

**Soundness:** 3
**Presentation:** 4
**Significance:** 3
**Originality:** 4
**Overall Recommendation:** 3
**Confidence:** 4

**Summary:**

This paper introduces AgentSteerTTS, a multi-agent closed-loop framework designed to improve expressive text-to-speech (TTS) systems when handling complex, composite emotional instructions (e.g., "Happy but slightly Arrogant"). To overcome the fundamental challenges of semantic-acoustic mismatch and speaker-emotion entanglement , the system utilizes three core components: an Adversarial Disentanglement Module (ADM) to prevent speaker-emotion feature leakage, a Dual-Stream Anchoring Controller (DAC) that grounds abstract text intents using concrete acoustic prototypes retrieved from a database, and a Fast-Slow Feedback Agent that dynamically refines the speech output during inference through both latent gradient correction and high-level perceptual critiques. Extensive experiments show that AgentSteerTTS significantly outperforms existing baselines by accurately rendering fine-grained composite emotions while preserving speaker identity and natural audio quality.

**Compliance With Llm Reviewing Policy:**

Affirmed.

**Key Questions For Authors:**

N/A

**Limitations:**

yes

**Strengths And Weaknesses:**

**Strengths**
1. The proposed AgentSteerTTS framework introduces a well-designed, closed-loop multi-agent system inspired by human cognitive decoupling
2. The rich visualizations provide excellent evidence that the model achieves true compositional control rather than just averaging features.

**Weaknesses**
1. This work heavily relies on the Prototype Database, which still needs to require manually text annotations.
2. The various MOS metrics (e.g., ESMOS, SNMOS, SSMOS, DMOS, .. etc) presented in the tables lack clear definitions or explanations in the text.
3. Why is there no direct comparison against the pure Gemini or Qwen3 models? It raises the question of whether the performance advantages simply come from the inherent capabilities of these underlying foundation models.
4. Based on Table 3, it appears that the performance improvement primarily comes from the "Prototype Retrieval (text-only)" component.
5. Insufficient Analysis of ADM Disentanglement.



I'm open to raise my score if the authors can solve these issues.

---

> ### Author Rebuttal · Authors · 2026-03-30
>
> We thank the reviewer for the thorough feedback.
>
> **Q1: Prototype Database.**
> Our method does not require dense prosody labels for the full training corpus. As described in **Sec.~3.2**, we build the prototype library $\mathcal{M} \)$ by curating 100 hours of high-expressivity segments from a 700-hour emotional speech pool. Each prototype is paired with an MLLM-initialized fine-grained description and then filtered by human listening to improve label quality and expressivity coverage, so the manual effort is mainly prototype-level screening/correction rather than dense annotation from scratch. This cost is justified by clear retrieval gains: as shown in **Appendix Table~4**, the full retrieval design reaches HumanMatch (=84\%) above text-only (70\%) and audio-only (66\%); and on the composite benchmark, anchored control improves E-SIM from 0.864 to 0.955 over the strongest baseline (**Table~2**).
>
> | Retrieval setting | R@1| R@5| Emo-SIM | HumanMatch |
> |--|--:|---:|--:|---:|
> | Text only | 0.61 | 0.88 |0.73 |70% |
> | Acoustic feature only | 0.56 | 0.85 | 0.75 | 66% |
> | Semantic + emo/style + penalty | 0.78 | 0.95 | 0.83 | 84% |
>
> **Q2: MOS unclear.**
> In the revision, we will explicitly define all MOS-style metrics at first use, state their rating targets, and unify the notation across **Sec.~4.1**, **Table~1**, and **Table~2**. .
>
> **Q3: Comparison with “pure Gemini/Qwen3”.**
> There is no pure Gemini/Qwen TTS baseline in **Tables~1--2**. The “Gemini3/Qwen3” entries denote the same AgentSteerTTS pipeline with different *external critic/judge backends* in the Slow Loop. This backend-swap experiment directly tests whether the gains come from the underlying LLM. With the same TTS backbone and controller, changing the judge reduces composite **E-SIM** from 0.955 to 0.921 $ \Delta = 0.034$, while **QMOS / SMOS / PMOS** change by at most 0.04 (**Table~2**). On ESD, WER/MCD change only slightly, and the corresponding subjective metrics in **Table~1** vary only marginally. This shows that the main gain comes from our decouple--anchor--closed-loop design, not from the judge itself.
>
> **Q4/Q5: Performance mainly from retrieval? ADM disentanglement analysis.**
> We agree that the current **Table~3** presentation can make the functional roles of different modules less explicit. In the revision, we will regroup the ablations by *function* and promote the key appendix evidence into the main paper.
>
> **(a) Retrieval is necessary for semantic anchoring.**
> Removing prototype retrieval (text-only control) drops **E-SIM** from 0.955 to 0.910 and **CSR** from 0.78 to 0.49 (**Table~3**), showing that discrete text alone often cannot reach the correct acoustic region. However, retrieval alone is still sensitive to noisy segments and low-confidence cases: our *perceptual cropping* and gated fusion reduce leakage from 0.21 to 0.14 and raise E-SIM to 0.955 (**Table~3**, **Fig.~5**). In the attribute-energy analysis (**Fig.~12**), the target-dimension share rises from 18.5% to 52.6%, non-target leakage drops from 81.5% to 47.4%, and temporal variance is reduced by 90%, which further shows that retrieval must be calibrated rather than used naively.
>
> **(b) ADM mainly improves identity stability and the disentanglement trade-off.**
> Removing ADM degrades **S-SIM** from 0.841 to 0.807 and increases **$\Delta$ S-SIM** from 0.021 to 0.048 (**Table~3**). More directly, **Fig.~10** shows that the correlation between emotion alignment and timbre drift drops from $\mathrm{Corr}(\mathrm{E\mbox{-}SIM}, \Delta\,\mathrm{S\mbox{-}SIM}) = 0.544$ without ADM to $-0.032$ with ADM ($\Delta r = 0.576$), while mean $\Delta$ S-SIM falls from 0.050 to 0.021 and E-SIM slightly improves ($0.949 \rightarrow 0.952$). Linear probes in **Fig.~11** also show reduced identity leakage: $z_{\mathrm{emo}} \rightarrow$ speaker drops from 97.5\% to 64.4\%, while $z_{\mathrm{emo}}$ still preserves emotion structure ($94.2\% \rightarrow 70.8\%$). These results directly support ADM’s role in improving the identity--emotion trade-off.
>
> **(c) The Fast--Slow loop mainly improves robustness and failure recovery.**
> Removing the Fast or Slow loop changes average E-SIM only slightly (0.955 $\rightarrow$ 0.951 / 0.949), but **CSR** drops to 0.70 / 0.67 (**Table~3**), showing that the loop acts mainly as a corrector rather than a source of raw alignment gain. This is consistent with **Fig.~6**, where feedback iterations improve control quality while remaining within the reported latency budget, and with **Fig.~5**, where adaptive gating gives its largest benefit on hard low-confidence cases.
>
> **Key ablation numbers (from Table~3).**
>
> | Setting | E-SIM | CSR | $ \Delta \$S-SIM  | Leak|
> |---|---:|---:|---:|---:|
> | Full | 0.955 | 0.78 | 0.021 | 0.14 |
> | w/o Retrieval | 0.910 | 0.49 | 0.025 | 0.18 |
> | w/o ADM | 0.948 | 0.61 | 0.048 | 0.17 |
> | w/o Fast | 0.951 | 0.70 | 0.023 | 0.15 |
> | w/o Slow | 0.949 | 0.67 | 0.022 | 0.16 |

---

> > ### Author Rebuttal · Reviewer_4Do5 · 2026-04-04
> >
> > Thank you for the clarification.

---

> > > ### Author Response · Authors · 2026-04-04
> > >
> > > Thank you again for your thoughtful review and follow-up response. We sincerely appreciate your recognition that our rebuttal has addressed your concerns. Your comments were very helpful in strengthening the paper, especially in clarifying the prototype database, the metric definitions, and the distinct roles of retrieval, ADM, and the feedback loops.
> > >
> > > We also hope our response has made clearer that the contribution of this work lies not only in strong objective and subjective performance, but also in its integrated decouple–anchor–closed-loop design, which consistently outperforms strong mainstream baselines and has already shown practical value in deployment across multiple games. If you feel that the main concerns have now been resolved, we would be very grateful if this could be reflected in your final assessment.
> > >
> > > Thank you again for your time, constructive feedback, and for helping improve the paper.

---

### Decision · Program_Chairs · 2026-04-30

**Decision:**

Accept (regular)

**Comment:**

This paper introduces AgentSteerTTS, a multi-agent closed-loop framework designed to address the challenge of fine-grained control for composite emotional instructions in expressive TTS systems. Reviewers acknowledged the timeliness of the problem, the novelty of applying a cognitive decoupling approach to TTS, and the strong empirical gains in composite emotion fidelity. However, some concerns were raised regarding the system's heavy reliance on a curated 100-hour acoustic prototype library, the fairness of comparing a retrieval-augmented model to zero-shot baselines, the limitations of a binary speaker-emotion disentanglement, and a regression in subjective naturalness scores (DMOS). In their rebuttal, the authors mitigated several of these issues by providing additional ablation studies. While the active reviewers acknowledged these clarifications and found them satisfactory, reservations linger regarding the reliance on a private, author-constructed benchmark for evaluation and the inherent structural bounds tied to the retrieval library's coverage. Balancing the practical significance of the task and the robust execution against these remaining architectural limitations, the paper clears the bar for publication and is recommended for a weak accept.